# Dispersion engineered metasurfaces for broadband, high-NA, high-efficiency, dual-polarization analog image processing

Michele Cotrufo [1,2,4] ✉, Akshaj Arora [1,3,4], Sahitya Singh[1,3] & Andrea Alù [1,3] ✉

Optical metasurfaces performing analog image processing – such as spatial differentiation and edge detection – hold the potential to reduce processing times and power consumption, while avoiding bulky 4 F lens systems. However, current designs have been suffering from trade-offs between spatial resolution, throughput, polarization asymmetry, operational bandwidth, and isotropy. Here, we show that dispersion engineering provides an elegant way to design metasurfaces where all these critical metrics are simultaneously optimized. We experimentally demonstrate silicon metasurfaces performing isotropic and dual-polarization edge detection, with numerical apertures above 0.35 and spectral bandwidths of 35 nm around 1500 nm. Moreover, we introduce quantitative metrics to assess the efficiency of these devices. Thanks to the low loss nature and dual-polarization response, our metasurfaces feature large throughput efficiencies, approaching the theoretical maximum for a given NA. Our results pave the way for low-loss, high-efficiency and broadband optical computing and image processing with free-space metasurfaces.

Image processing plays a key role in rapidly advancing technologies such as augmented reality, advanced driver assistance systems, and biomedical imaging, and it is typically performed digitally. Despite their versatility, digital approaches are affected by several drawbacks, such as low operational speed and energy consumption, which are critical factors in several applications. These limitations can be overcome by developing analog components, optimized to perform specific calculations, which run in parallel to digital processors in order to enhance the overall speed and efficiency. Optics is the foremost approach for implementing such analog computations, due to the unmatched speed, low power consumption[1,2] and ease of re-configurability into various network topologies[3–7] For example, analog image processing is commonly performed via Fourier filtering techniques[8] whereby a 4F lens system is used to physically access the Fourier transform of an input image and filter different frequency components via spatially selective masks. Despite its simplicity, the required 4F configuration makes this approach inherently bulky and prone to alignment issues, and thus not well-suited for integrated

devices. In recent years, optical computing[9,10] has garnered renewed interest because the current nanotechnology tools have enabled the development of artificially engineered materials that allow to increase complexity[11] while maintaining compact footprints. In this context, metasurfaces – planarized ultra-thin artificial devices - have already been successfully employed for several tasks such as light wavefront shaping[12], thin polarization optics[13], and compact spectrometers[14]

Recently, it has been proposed[15,16] that metasurfaces can be also used as compact devices to implement analog image processing without requiring to physically access the Fourier space, as schematized in Fig. 1a. Assume that an optical image is defined in the plane $z = 0$ by an intensity profile $I_{in}(x,y) = |\mathbf{E}_{in}(x,y)|^2$, where $\mathbf{E}_{in}(x,y) = E_{in}(x,y)\mathbf{e}$ is an electric field with polarization direction $\mathbf{e}$ and angular frequency $\omega = 2\pi c/\lambda = k_0 c$. Following standard Fourier optics[17], the image can be decomposed into a bundle of plane waves, each propagating along a direction identified by the polar and

[1]Photonics Initiative, Advanced Science Research Center, City University of New York, New York, NY 10031, USA. [2]The Institute of Optics, University of Rochester, Rochester, NY 14627, USA. [3]Physics Program, Graduate Center of the City University of New York, New York, NY 10016, USA. [4]These authors contributed equally: Michele Cotrufo, Akshaj Arora. ✉e-mail: mcotrufo@optics.rochester.edu; aalu@gc.cuny.edu

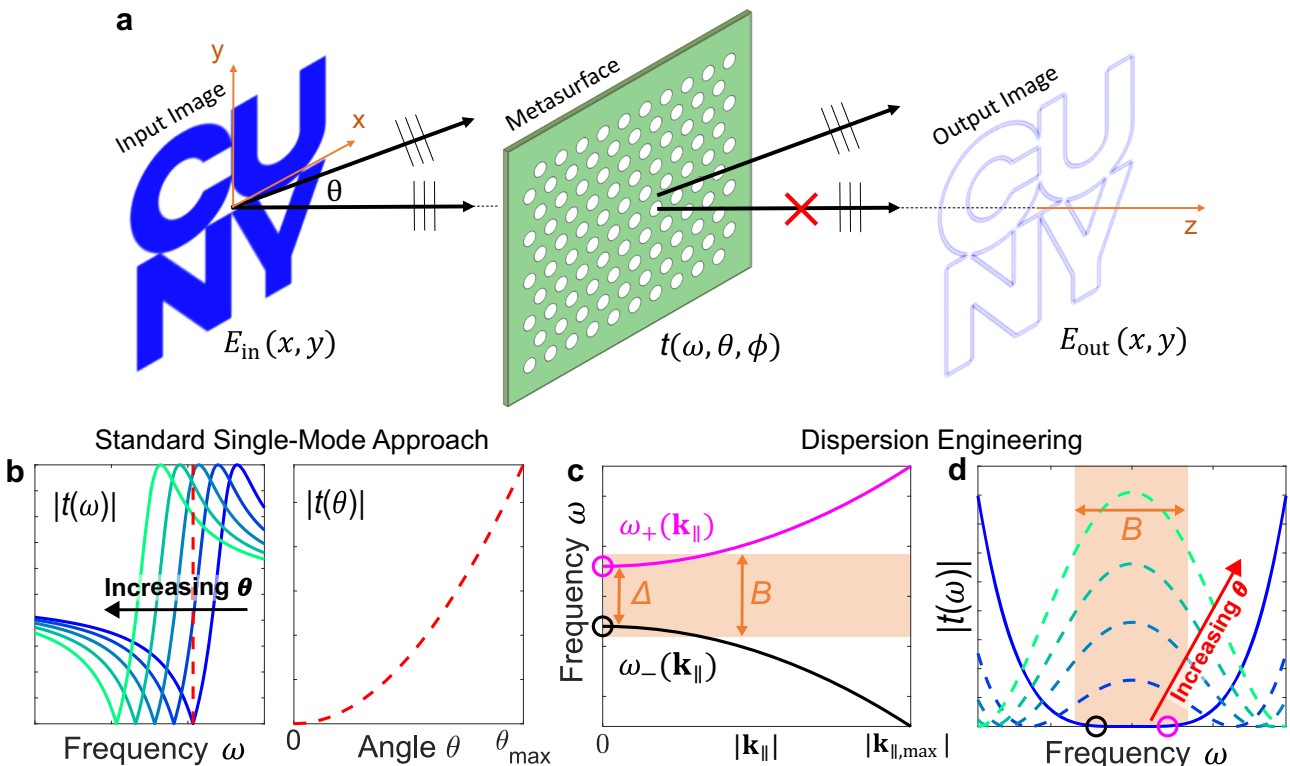

**Fig. 1 | Two-dimensional image differentiation using metasurfaces. a** Schematic of using a metasurface to implement the Laplacian operation on an input image. **b** Sketch of the conventional approach to achieve the Laplacian response at a single frequency. Left: the transmission amplitude $|t(\omega)|$ of a single-mode Fano lineshape spectrally shifts as a function of the incident angle. Right: at a single frequency (marked by the vertical dashed line in left part), the transmission amplitude versus angle ($|t(\theta)|$) displays the required Laplacian behavior. **c, d** Schematic of the concept demonstrated in this work. **c** Targeted dispersion relation, consisting of two modes $\omega_+(\mathbf{k}_\parallel)$ and $\omega_-(\mathbf{k}_\parallel)$ whose frequencies shift towards opposite directions as the in-plane wave-vector increases. **d** Resulting angle-dependent transmission amplitude, displaying a large bandwidth $B$ (shaded area) with almost-zero transmission at normal incidence.

azimuthal angles $\theta$ and $\phi$ and with amplitude proportional to the Fourier transform $f_{in}(k_x, k_y) = \int dx\, dy\, e^{-i(k_x x + k_y y)} E_{in}(x, y)$, where $[k_x, k_y] = k_0 \sin\theta [\cos\phi, \sin\phi]$. In a standard imaging setup, these plane waves are collected and re-focused by a pair of lenses or objectives (not shown in Fig. 1a), effectively performing an inverse Fourier transform and re-creating the image at a different plane. Thus, any mathematical operation defined in the Fourier space can be performed by selectively filtering the bundle of plane waves originating from the image with respect to their propagation direction. This angle-selective filtering – typically hard to control with naturally available materials - can be implemented with suitably designed metasurfaces. By tailoring the scattering properties of the metasurface, the desired mathematical operation can be encoded into the metasurface transfer function.

An analog operation that has received much attention in the past years is the edge detection[18–29] whereby the edges of an input image $E_{in}(x, y)$ are enhanced with respect to homogeneous regions (Fig. 1a). This can be obtained by applying a spatial differential operator, such as the Laplacian operator $E_{out}(x, y) = (\partial_x^2 + \partial_y^2) E_{in}(x, y)$, which translates, in Fourier space, into a high-pass filter described by $f_{out}(k_x, k_y) = -(k_x^2 + k_y^2) f_{in}(k_x, k_y)$. Qualitatively, this operation can be obtained by a metasurface that suppresses plane waves propagating at small angles ($\theta \approx 0$), while progressively transmitting waves propagating at larger angles (Fig. 1a). In order to perform this operation on an arbitrarily polarized input 2D image, a metasurface must feature, for any azimuthal angle and for any polar angle below a certain

maximum value $\theta \leq \theta_{max}$, a polarization-independent and isotropic transfer function of the form[19]

$$t(\theta, \phi) = \begin{pmatrix} t_{ss}(\theta, \phi) & t_{sp}(\theta, \phi) \\ t_{ps}(\theta, \phi) & t_{pp}(\theta, \phi) \end{pmatrix} = \begin{pmatrix} C\sin^2\theta & 0 \\ 0 & C\sin^2\theta \end{pmatrix} \quad (1)$$

where $C$ is an overall pre-factor and the subscripts denote s and p polarization (see also mathematical derivation in Supplementary Information, Supplementary Section S3). While in numerical experiments any transfer function described by Eq. 1 leads to second-order differentiation of a certain class of input images, in practical applications several figures of merit and key metrics need to be optimized to guarantee acceptable performance. To be able to process high-resolution images without image distortion, the metasurface numerical aperture $NA \equiv \sin(\theta_{max})$ must satisfy $NA \geq k_{in, max}/k_0$, where $k_{in, max}$ is the largest wave vector in the Fourier decomposition of the input image. Thus, the ideal behavior described by Eq. 1 must remain valid up to very large polar angles. Moreover, low values of $|C| \ll 1$ are detrimental because they strongly reduce the intensity of the processed image, resulting in large inefficiencies. Finally, while responses similar to Eq. 1 can be readily engineered at a given frequency, in many practical applications the input light might either have an unknown frequency or be polychromatic, and with a broad spectrum. Thus, it is highly desirable to use a metasurface for which the transfer function in Eq. 1 is maintained over a broad range of frequencies.

Several theoretical[18–22] and experimental[23–29] works have discussed different approaches to perform edge detection with metasurfaces

under certain excitation/collection conditions. A common approach[19–21,24,28] relies on inducing a single optical mode in a nonlocal metasurface, leading to a Fano lineshape in the normal-incidence transmission spectrum, as sketched in Fig. 1b (left plot). As a result, the normal-incidence transmission is zero at a single operational frequency ω (red dashed line in left plot of Fig. 1b). As the angle θ increases, the Fano lineshape spectrally shifts due to the metasurface angular dispersion, leading to the desired increase of the transmission at the fixed frequency ω, as sketched in the right plot of Fig. 1b. While this approach has proven successful in realizing analog edge detection, its operational bandwidth is inherently limited. Indeed, the almost-zero transmission at normal incidence, necessary to suppress the low-spatial-frequency components, can be obtained only in a range of frequencies much smaller than the linewidth of the Fano resonance. The operational bandwidth can be increased by employing an optical mode with lower Q factor, in order to obtain a wider band of almost-zero transmission at normal incidence[28]. However, modes with larger linewidths also lead to a slower increase of the transmission as the angle θ increases. This reduces the transmission of the high-spatial-frequency components, effectively establishing a trade-off, common to most of the current literature, between spectral bandwidth and intensity throughput. Moreover, the nonlocal metasurfaces demonstrated experimentally so far feature either a $C_2$[24] or $C_4$[28] rotational symmetry, which introduces large azimuthal anisotropies and/or strong polarization asymmetries. Approaches based on topological photonics[27,29] can provide broader spectral bandwidths and better isotropy, but they typically require working in a cross-polarized reflection modality and at well-defined and large angles of incidence, which sets hard lower bounds on the footprint of the overall device and strongly limits the numerical aperture and practical operation. A largely different class of edge-detection devices has been recently proposed by placing Pancharatnam-Berry phase metasurfaces in the Fourier plane of a 4F system combined with two crossed polarizers[30–32] This approach typically leads to large operational spectral bandwidths, since working in a 4F modality allows to effectively decouple the spectral and angular response of the metasurface. However, the 4F arrangement and the required polarizers strongly constrain the minimum footprint of this approach, and thus limit the possibility of miniaturization. We note that, as for any analog optical filtering technique that relies on coherent superposition of different plane waves, this approach to edge detection requires some degree of spatial coherence in the illumination. Recent works have proposed a way to overcome this limitation[33,34] albeit at the expense of increased footprint and the need of digital postprocessing.

Overall, despite a few implementations discussed in the literature, a metasurface that can perform isotropic, broadband, polarization-independent, high-NA, and high-efficiency edge detection without using a 4F arrangement, while keeping the footprint compact and the design feasible for fabrication, has not been demonstrated yet. In this context, it is worth emphasizing that image-processing approaches based on using a spatially-varying metasurface as a mask in a 4F system[26,30–32] offer very limited possibilities in terms of miniaturization, since the footprint of the overall setup is largely dominated by the lens thicknesses and focal lengths. Moreover, it is important to stress that the efficiency of the edge-detection process, i.e., how the intensity of the output image compares to the intensity of the input image, has been overlooked in recent works, yet it is of paramount importance for practical applications.

In this work, we propose and experimentally demonstrate a route based on dispersion engineering to realize edge-detection metasurfaces with optimal figures of merit discussed above. We showcase the potential of our design principle in a silicon-on-glass platform, demonstrating designs with fully isotropic responses, NAs larger than 0.35, operational bandwidths of 35 nm around a central wavelength of about 1500 nm, and operation for any input polarization – important

requirements to facilitate the adoption of these devices in real-world application. In order to quantify the edge-detection performance, we introduce two metrics to assess the efficiency of our devices, quantifying in a more rigorous way the throughput and insertion loss of the metasurface –important benchmarks when comparing the performance of different designs. Importantly, we demonstrate that the efficiency experimentally achieved in our work is close to the efficiency of any ideal k-space filter with the same NA. Moreover, the bandwidth demonstrated here can be largely increased by further engineering the metasurface dispersion. As an example, in the Supplementary Information (Supplementary Section S6) we numerically demonstrate a design where the bandwidth is increased two-fold, albeit at the expense of a slightly increased background.

## Results
### General principle and metasurface design
We first introduce the general concept behind our approach, and later discuss its implementation in a realistic design. Our recipe to obtain broadband, high-efficiency, and high-NA edge-detection relies on engineering the band structure dispersion of a periodic nonlocal metasurface. Specifically, we engineer two different dispersive modes, denoted $\omega_+(\mathbf{k}_\parallel)$ and $\omega_-(\mathbf{k}_\parallel)$ (Fig. 1c), such that: (I) the frequency of the two modes are different but close at the Γ point ($\mathbf{k}_\parallel = 0$), with $\Delta \equiv \omega_+(0) - \omega_-(0) > 0$, (II) the radiative linewidths of the modes $\omega_+(0)$ and $\omega_-(0)$ are comparable to or slightly larger than their detuning $\Delta$, and (III) the two modes shift spectrally towards different directions as $|\mathbf{k}_\parallel|$ increases, with the frequency $\omega_+(\mathbf{k}_\parallel)$ increasing and the frequency $\omega_-(\mathbf{k}_\parallel)$ decreasing. This behavior, schematically depicted in Fig. 1c, can be obtained for example at photonic crystal bandgaps. As shown schematically in Fig. 1d, conditions (I) and (II) result into a large band $B > \Delta$ of almost-zero transmission at normal incidence (solid blue line and orange-shaded area in Fig. 1d). The spectral width of this band is approximately double the linewidth of the two modes. Furthermore, due to condition (III), as the angle θ increases the transmission increases at all frequencies within the band B. Moreover, by ensuring that the dispersion of the two modes is strong enough that their overall spectral shifts are larger than their linewidths, the transmission within the band $B$ will rise to almost-unitary values for large angles θ. This property ensures high transmission of large spatial-frequency Fourier components, resulting in higher intensities in the output images. Finally, while the previous guidelines can be applied to any periodic metasurfaces with arbitrary lattice symmetry, working with lattices with $C_6$ rotational symmetry leads to a polarization-independent response at normal-incidence, and it guarantees the largest possible degree of isotropy for tilted angles. We emphasize, however, that a $C_6$ rotational symmetry does not automatically guarantee polarization-independent response at off-normal angles, which is instead crucial to achieve uniform edge detection independently of the polarization of the input image. Achieving a full-angle polarization-independent response requires to further engineer the dispersion of the metasurface to ensure that the relevant optical modes have a similar coupling to s- and p-polarized waves.

Remarkably, we show that all these seemingly stringent requirements can be achieved within a relatively simple metasurface platform composed of a photonic crystal slab over a transparent substrate (Fig. 2). In this work we consider silicon metasurfaces operating in the near-infrared (NIR), but the concept can be readily generalized to any spectral region. Figure 2a shows the unit cell of the proposed design, which consists of a triangular lattice of air holes etched into a silicon slab (relative permittivity $\varepsilon_r = 11.90$) and placed on glass ($\varepsilon_r = 2.31$) for mechanical support. The device design is fully characterized by three parameters: the lattice constant $a$, the thickness $H$ and the radius of the holes $R$. We fix the lattice constant to $a = 924$ nm in order to achieve operational wavelengths close to 1500 nm, and we vary the slab thickness $H$ and the hole radius $R$ to optimize the band structure and

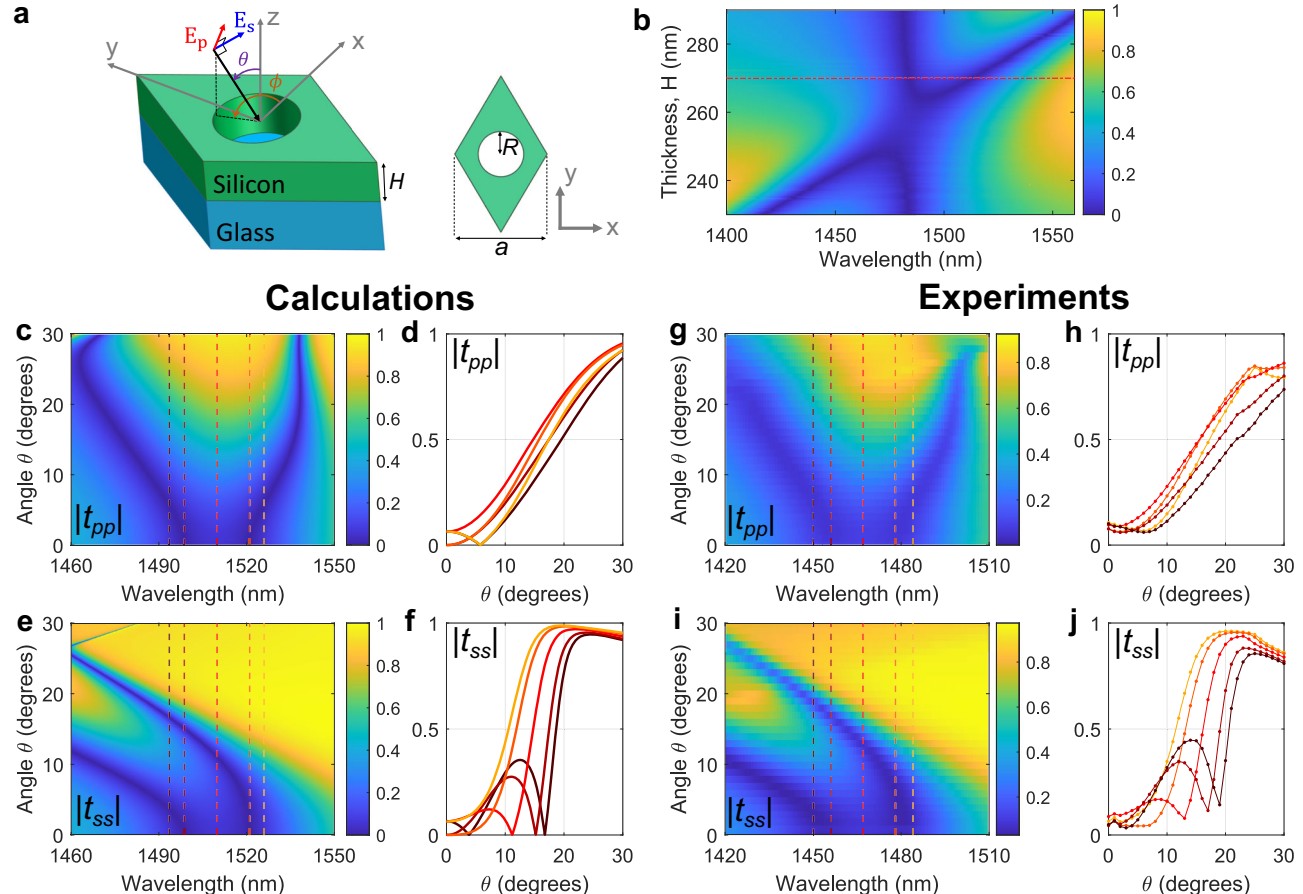

## Calculations

## Experiments

**Fig. 2 | Geometry, numerical simulations, and experimental measurements.**
**a** Unit cell of the metasurface considered in this work. All devices considered in this work have lattice constant $a = 924$ nm. **b** Numerically calculated normal-incidence transmission spectra for devices with fixed radius $R = 265$ nm and varying thickness $H$. **c** Numerically calculated magnitude of the p-polarized transmission coefficients, $|t_{pp}|$, versus the incident angle $\theta$ and wavelength, for a device with $R = 265$ nm and

$H = 270$ nm (red dashed-dotted line in panel **b**) and for $\phi = 0$. **d** Calculated $|t_{pp}|$ versus $\theta$ for five increasing wavelengths $\lambda_1, \lambda_2, \lambda_3, \lambda_4, \lambda_5$, denoted by color-coded vertical lines in panel **c**. **e**, **f** Same as panels **c**, **d**, but for the s-polarized transmission coefficient $|t_{ss}|$. **g–j** Same as panels **c–f**, but experimentally measured (see text and Supplementary Information for additional details).

meet the three conditions described above. All simulations were performed numerically with a commercially available software (Ansys HFSS). Figure 2b shows the normal-incidence transmission amplitude versus impinging wavelength, for fixed $R = 265$ nm and for various thicknesses $H$. The transmission spectra are dominated by two modes, whose distance can be readily controlled by $H$. Following conditions I and II described above, we select $H = 270$ nm (dashed-dotted line in Fig. 2b), where the spectral detuning of the two modes is comparable to their linewidths. Figure 2c shows, for this optimized device, the simulated p-polarized transmission amplitude $|t_{pp}|$ versus wavelength and for increasing values of the polar angle $\theta$. As clear from the color plot, the two modes follow the targeted behavior (see also Fig. 1c, d) diverging from each other up to large angles $\theta_{max} \approx 26°$. This feature results in a bandwidth of about 35 nm (5 THz), centered approximately around 1500 nm, over which the desired Laplacian-like transfer function is obtained. The spectral bandwidth could be increased by working with optical modes that are further detuned at normal incidence, albeit at the expense of a larger background at the central frequency. As an example, in the Supplementary Information (Supplementary Section S6) we numerically demonstrate a design with slightly different values of $R$ and $H$, which features a bandwidth of 10 THz.

The quality of the Laplacian-like response is further highlighted in Fig. 2d, where we show vertical cut lines at selected wavelengths corresponding to the color-coded vertical lines in Fig. 2c. For all

wavelengths, $|t_{pp}(\theta)|$ increases almost monotonically with $\theta$, reaching values above 0.9 for $\theta \approx 30°$. The s-polarized transmission amplitude, shown in Fig. 2e, f, displays a similar behavior although with some nonidealities. At $\theta = 0°$ the two transmission zeros occur at the same wavelengths as for p-polarization, due to the triangular lattice symmetry. For $\theta > 0°$, the lower-wavelength mode shifts towards shorter wavelengths, as required. The higher-wavelength mode, instead, is almost dispersionless for small values of $\theta$ and it shifts towards shorter wavelengths as $\theta$ increases. Therefore, the associated transmission zero crosses through the operational band. As a result, for certain wavelengths the transmission amplitude does not increase monotonously with $\theta$, but it instead features a zero for intermediate values of $10° < \theta < 20°$ (Fig. 2f). Despite this potential issue, we note that all transfer functions in Fig. 2f display the desired high-pass-filter behavior, suppressing low spatial frequencies while promoting high frequencies. As we demonstrate experimentally in the following, this sub-ideal behavior for s-polarization does not introduce any practical detrimental effect in the edge-detection functionality. Moreover, the transmission amplitudes reach values of almost 1 for $\theta \approx 20°$ (Fig. 2f), and they remain almost flat for larger angles, further highlighting the large efficiency and NA of this device.

In order to verify the edge-detection performance of this metasurface, and in particular its isotropy, we calculated the angle-dependent transfer functions $t_{pp}(k_x, k_y)$ and $t_{ss}(k_x, k_y)$ for each of the five wavelengths considered in Fig. 2d, f. The amplitudes of the transfer

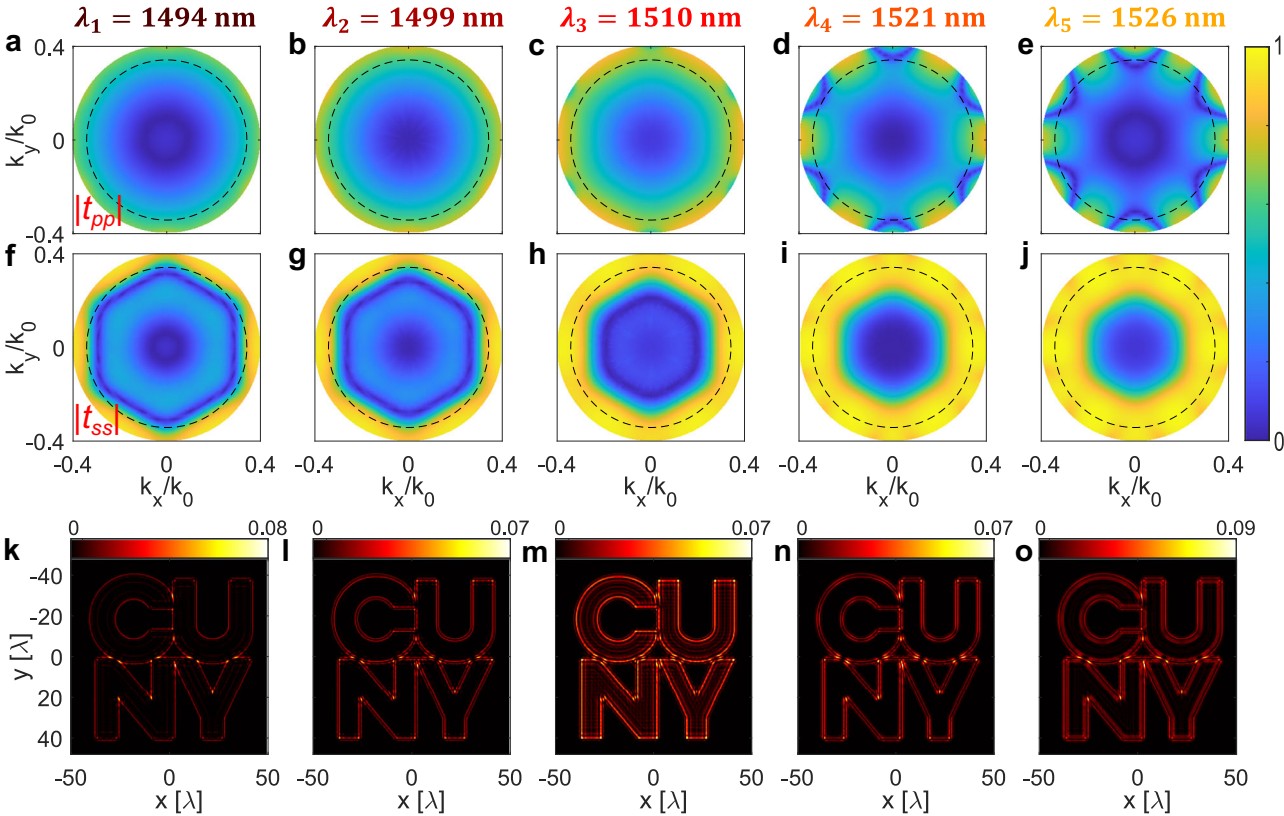

**Fig. 3 | Numerical calculations of transfer functions and edge detection.**
**a−e** Absolute value of the p-polarized transfer function of the device considered in
Fig. 2c−f, versus the in-plane wave-vector and for the five different wavelengths
$\lambda_1 - \lambda_5$ considered in Fig. 2c−f, as indicated above each panel. The dashed circles
denote NA = 0.35. **f−j** Same as for panels **a−e**, but for the s-polarized transfer
function. **k−o** Calculated output images for each wavelength and for unpolarized

excitation. The input image is the CUNY logo shown in Fig. 1a, where the high-
intensity areas (the inner part of the letters) have intensity equal to 1, while the
lower-intensity areas have intensity equal to zero. For each of the panels **k−o**, the
efficiency $\eta_{\text{peak}}$ (defined in the text) can be obtained from the upper limit of the
corresponding colorbar.

functions (Fig. 3a−e for p-polarization and Fig. 3f−j for s-polarization)
show that, up to $\theta \approx 21°$ (NA = 0.35, dashed circles in the transfer
functions in Fig. 3), the device response is almost fully isotropic with
respect to the azimuthal angle $\phi$. The phases of the transmission
amplitudes, shown in the Supplementary Information (Supplementary
Section S2.1), are almost independent of $\theta$ and $\phi$, as required for a
proper implementation of the Laplacian operation. The cross-
polarized transfer functions, $t_{sp}(k_x,k_y)$ and $t_{ps}(k_x,k_y)$, shown in the
Supplementary Information (Supplementary Section S2.1), have van-
ishing amplitudes up to NA = 0.35. In order to verify the edge detection
functionality, we numerically calculated the output images created by
this metasurface for different input wavelengths, using the CUNY logo
shown in Fig. 1a as the input image. The pixel size of the input image is
rescaled such that the lateral extent of the logo is equal to about $80\lambda$
($\sim 120$ μm). In order to readily estimate the edge detection efficiency
(see discussion below), we set the peak intensity of the input image
equal to 1. To obtain the output image (see Supplementary Informa-
tion, Supplementary Section S3 for the detailed derivation), we cal-
culate the full polarization-dependent plane-wave expansion of the
input image, and then calculate the filtering effect of the metasurface
by including both co-polarized ($t_{ss}$ and $t_{pp}$) and cross-polarized
($t_{sp}$ and $t_{ps}$) complex transfer functions. Moreover, we assume that the
input image is created by an unpolarized electric field. The output
images in Fig. 3k−o confirm that the metasurface filtering results into
sharp edges and a suppression of the homogeneous background at
multiple wavelengths (each plot is calculated for the wavelength
reported on top the corresponding column). Thanks to the metasur-
face isotropy, the intensity and appearance of the edges are quite

uniform and independent of the edge direction. In order to quantita-
tively assess the efficiency of our edge-detection metasurface, it is
necessary to quantify how the intensity of the filtered image compares
with the intensity of the input image. This metric is of crucial impor-
tance for the implementation of these devices in real-world appli-
cations because it dictates, for example, how much the integration
time of a camera needs to be increased in order to have the same
signal level in the output and input image. To this aim, we introduce
two different metrics. First, we consider the *peak efficiency*, defined
as the ratio $\eta_{\text{peak}} \equiv \max(I_{\text{out}}) / \max(I_{\text{in}})$ between the peak intensities in
the output and input images. The advantage of this metric is that it
can be easily estimated. Since we have set $\max(I_{\text{in}}) = 1$, the efficiency
$\eta_{\text{peak}}$ is equal to the upper limit of each colorbar in Fig. 3k−o, and it
ranges between 7% and 9%. However, due to its definition, the value
of $\eta_{\text{peak}}$ is typically determined by the highest-intensity pixels in the
images (corresponding to small regions with very large derivatives),
and it cannot correctly quantify the global efficiency. We therefore
introduce also the *average efficiency* $\eta_{\text{avg}} \equiv \text{avg}(I_{\text{out}}^{\text{edge}}) / \max(I_{\text{in}})$,
where $\text{avg}(I_{\text{out}}^{\text{edge}})$ is the average intensity of the output image calcu-
lated only in narrow regions surrounding the expected positions of
all edges. For the images in Fig. 3k−o, $\eta_{\text{avg}}$ varies between 1.5% and 3%
depending on the wavelength. As we demonstrate in the Supple-
mentary Information, these efficiencies are fairly close to the max-
imum theoretical value obtainable with any ideal edge-detecting
device with the same NA. We emphasize that such high efficiencies
are achieved due to the fact that our metasurface provides a
Laplacian-like response for both polarizations, and with large values
of transmission amplitudes at large angles.

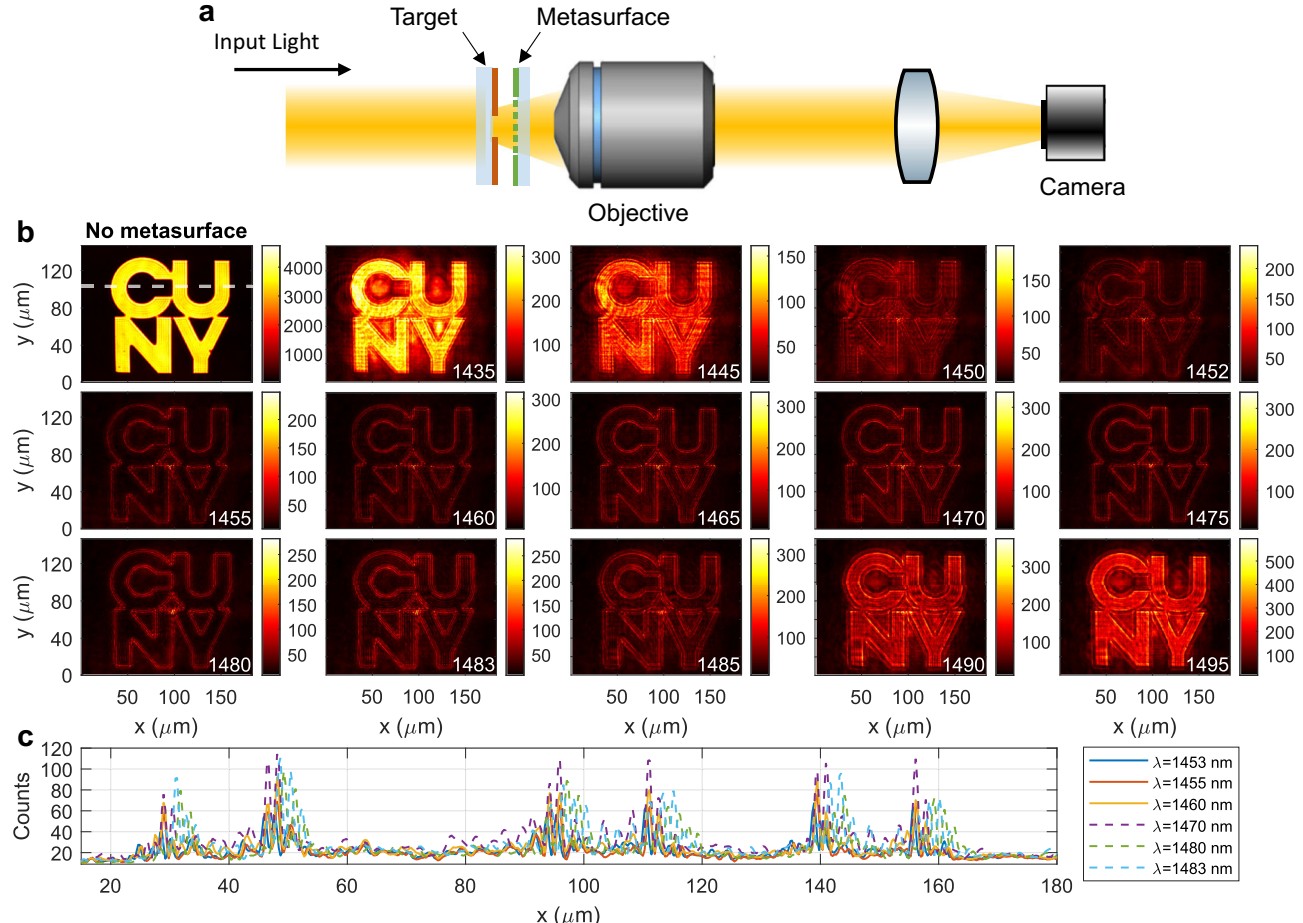

**Fig. 4 | Experimental edge detection with unpolarized narrow-band input.**
**a** Schematic of the setup used for the imaging experiments. **b** Top-left plot: Unfiltered image, obtained by removing the metasurface from the setup in panel **a**. All other colorplots show the output images obtained when the metasurface is placed in front of the target, for different impinging wavelengths (reported at the bottom-right corner of each plot). **c** Horizontal cuts of selected wavelengths from panel **b** (as indicated in the legend), corresponding to the vertical position denoted by the white dashed line in the top-left plot of panel **b**. For all measurements (with and without metasurface), the counts recorded by the camera have been normalized by the camera integration time and the power impinging on the target. The illumination is almost unpolarized (measured degree of polarization ≈ 10%).

## Experimental verification

After having numerically verified the performance of the proposed metasurface design, we now experimentally demonstrate its operation. We fabricated the metasurface in Figs. 2, 3 by using amorphous silicon deposited on a glass substrate (see "Methods" section for details on fabrication), and we measured its transfer functions with a custom-built setup (see "Methods" section for details on the setup). The experimental measurements are shown in Fig. 2g–j side-by-side to the corresponding simulated results, and they show excellent agreement with the calculations. Specifically, for p-polarized excitation the transmission amplitude versus wavelength and $\theta$ (Fig. 2g) clearly shows the occurrence of the two transmission zeros at normal incidence, and their diverging spectral shift for increasing values of $\theta$. Similarly, the measured s-polarized transmission amplitude (Fig. 2i, j) agrees well with the simulated plots (Fig. 2e, f), and it displays the expected global increase of the transmission amplitude versus $\theta$ (Fig. 2j). All measurements and simulations in Fig. 2 are performed for a fixed value of the azimuthal angle $\phi = 0$. Additional experimental data for $\phi = 30°$ are available in the Supplementary Information (Supplementary Section S2.2).

Next, we demonstrate experimentally the edge-detection functionality of our device. To create the input image, we used a target with the shape of our institution logo, obtained by depositing a 200-nm-thick layer of chromium on a glass substrate and then etching the

desired shape after an electron beam lithography process. In our setup (see Fig. 4a and additional details in Supplementary Information, Supplementary Section S1), the target is illuminated by a collimated beam, which acts as a wide-field illumination, and the image scattered by the target is collected by a NIR objective (Mitutoyo, 50X, NA = 0.42) and relayed on a near-infrared camera with a tube lens with a $f = 15$ cm focal length. To perform edge detection, the metasurface is placed between the objective and the target, at a distance of few hundreds microns from the target. In order to correctly quantify the efficiencies $\eta_{peak}$ and $\eta_{avg}$, in all measurements (with and without the metasurface) we normalize the counts read by the camera by the camera integration time and the power impinging on the target.

We performed a first experiment with almost-unpolarized (degree of polarization < 10%) and narrowband (FWHM ≈ 5 nm) illumination, which is obtained by filtering the output of a supercontinuum laser with a commercial tunable filter. Figure 4b shows the output image obtained without the metasurface (top-left plot), together with several output images obtained with the metasurface in front of the target and at different input wavelengths, swept across the operational bandwidth determined by Fig. 2g–j. The illumination wavelength (in nanometers) is reported in the bottom-right corner of each color plot of Fig. 4b. As clear from these measurements, high-quality edge detection can be obtained for any wavelength between 1452 nm and 1485 nm. The extent of this bandwidth matches very well the simulations in

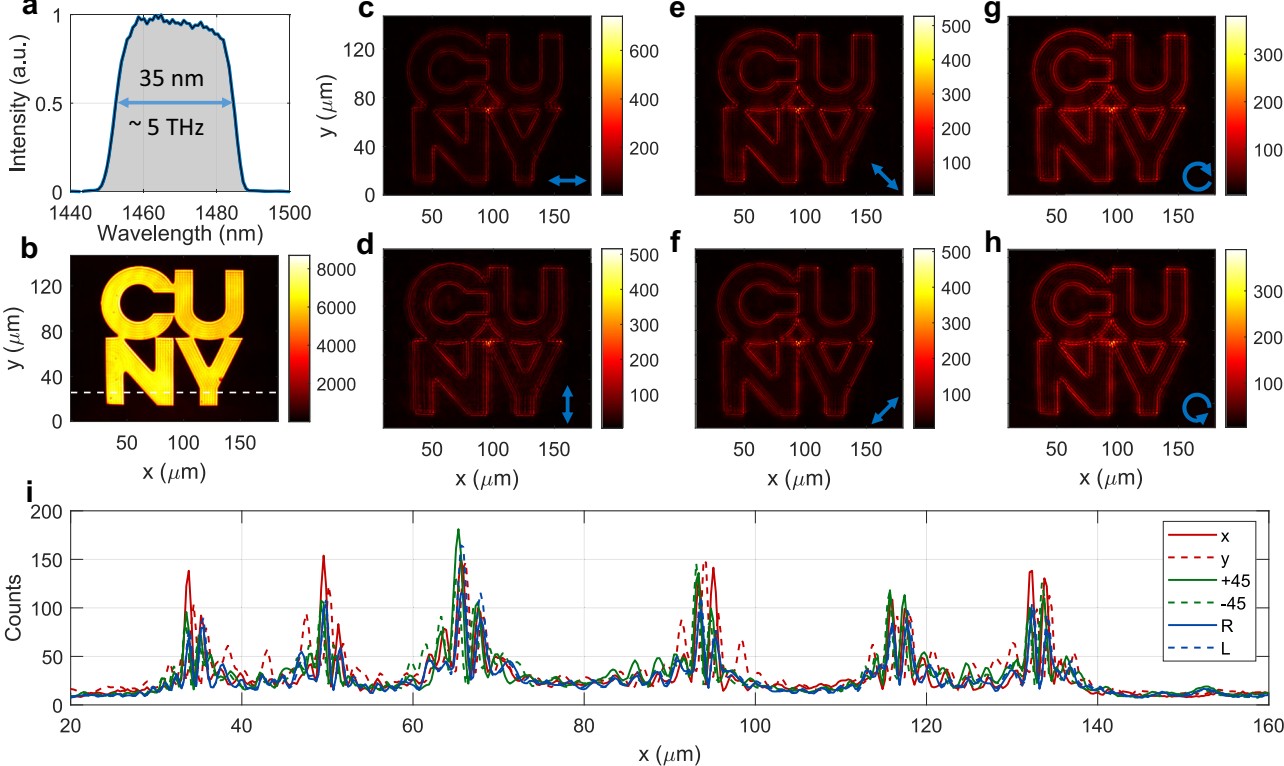

**Fig. 5 | Experimental edge detection with polarized broad-band input.**
**a** Spectrum of the excitation. **b** Unfiltered image. **c–h** Output images when the metasurface is placed in front of the target and for six different polarizations of the input light: linearly polarized along (**c**) x, (**d**) y, (**e**) the x-y diagonal, (**f**) the x-y anti-diagonal, or (**g**) right and (**h**) left circularly polarized. **i** Horizontal cuts of the plots in panels **c–h** (as indicated in the legend), corresponding to the vertical position denoted by the white dashed line in panel **b**. For all measurements (with and without metasurface), the counts recorded by the camera have been normalized by the camera integration time and the power impinging on the target.

Fig. 3. Moreover, for all wavelengths between 1452 nm and 1485 nm the efficiencies $\eta_{peak}$ and $\eta_{avg}$ reach values >5% and >1%, respectively, experimentally confirming that our metasurface can perform broad-bandwidth edge detection while maintaining a very large throughput. As mentioned in the previous section, values of $\eta_{peak} \geq 5\%$ are very close to the peak efficiencies obtainable with an ideal k-space filter performing the Laplacian operator with a fixed NA = 0.35 ($\eta_{peak}^{(ideal)} = 8\% - 9\%$, see also Supplementary Information, Supplementary Section S4). Thus, the experimental efficiency of our metasurface is very close to the upper theoretical bound. The edge-detection performance is further confirmed by the horizontal-cut intensity distributions shown in Fig. 4c (corresponding to the white dashed line in top-left plot of Fig. 4b). High-intensity peaks at the expected edge positions are obtained, surrounded by an almost-zero and uniform background. As expected from the second-order differentiation of a step-like function, each edge results in a pair of high-intensity peaks in the output images. From a practical point of view, the presence of these two peaks (as opposed to a single peak) can be useful to better identify the spatial position of the detected edge. Besides the two high-intensity peaks, some additional weaker peaks and noise are visible in the output images (see also Fig. 4c). As explained in more detail in the Supplementary Information (Supplementary Section S5), the additional weaker peaks are due to the fact that, in the input image, the intensity is not exactly constant inside (or outside) the CUNY logo. Instead, weak intensity fluctuations are present due to diffraction at the metallic aperture used to create the input image (see also Fig. S8 in Supplementary Information). Since our metasurface performs high-efficiency differentiation on the whole input image, these spatial intensity fluctuations in the input image will be differentiated as well, resulting in additional sets of weaker peaks in the output image. From a

practical point of view, the presence of these additional weaker peaks does not limit the capability of detecting the position of the main edges of the image, because the intensity of each peak in the output image remains proportional to the square of the spatial derivative (at the same position) of the input image. In our experiment, the peaks in the output images corresponding to the main edges are about 4x more intense than the peaks corresponding to the weak intensity fluctuations.

Finally, we experimentally verify the quality of the edge detection under a broad-band input. A broad excitation spectrum, extending from 1450 nm to 1485 nm (Fig. 5a), was obtained by filtering the output of a supercontinuum laser with a custom-built pulse shaper. Figure 5b shows the unfiltered image (i.e., without the metasurface), while Fig. 5c–h show the filtered image for different polarizations of the input illumination (as described in the figure caption). Our experimental results confirm that the output images feature very sharp and high-contrast edges also for very broad excitations. Remarkably, these measurements confirm that the quality and uniformity of the edges is essentially independent of the input polarization. This confirms that, despite the sub-ideal features for s-polarized excitation (Fig. 2e, i), our metasurface is still capable of performing high-quality edge detection for any input polarization. Moreover, even for broadband and arbitrarily polarized input, the efficiency remains quite high, with $\eta_{peak} \geq 3.5\%$ and $\eta_{avg} \approx 1\%$ for all images in Fig. 5c–h. The quality of the edges and the clear background suppression can also be appreciated by the horizontal cuts in Fig. 5i, which show the intensities of the filtered images (at the location marked by the horizontal white line in Fig. 5b) for the six different polarizations. In the Supplementary Information (Supplementary Section S2.3) we provide similar measurements for other target shapes.

## Discussion

In this work, we have theoretically proposed and experimentally demonstrated an approach to design metasurfaces performing analog edge detection over a broad bandwidth of input frequencies and for arbitrary polarization inputs, while simultaneously maintaining high efficiency, high NA, and excellent isotropy. Differently from conventional methods that rely on the angle-dependent spectral shift of a single optical mode, our approach relies on accurately engineering the angular dispersion of two different modes in a transversely invariant nonlocal metasurface. We have shown that this method provides enough flexibility and degrees of freedom to simultaneously optimize all the relevant figures of merit. Despite its apparent complexity, this approach can be readily implemented in different platforms and materials. In particular, we have experimentally implemented our approach in a silicon-on-glass platform, demonstrating a simple single-layer metasurface with isotropic responses up to large numerical apertures (NA > 0.35), and with an operational bandwidth of 35 nm (5 THz) around a central wavelength of ~1500 nm. We experimentally verified that this device performs high-quality and high-efficiency second-order differentiation for any input polarization state and for any wavelength within the bandwidth. Moreover, in order to fully and quantitively assess the performance of the device, we introduced two different efficiency metrics to quantify how the peak and average intensity of the edges compares to the intensity of the input image. Thanks to its dual-polarization response and to the large transmission values achieved at large angles, our device can achieve very large efficiencies for both monochromatic and broadband inputs, and for any input polarization. In particular, the experimentally measured efficiencies demonstrated in this work are very close to the maximum theoretical efficiency achievable with a passive k-space filter with a given NA, and they are only a factor of ~2x smaller than the efficiency that would be obtained by applying the exact Laplacian operation to the input images (see additional discussion in Supplementary Information, Supplementary Section S4).

We emphasize that the spectral bandwidth demonstrated here (35 nm around 1470 nm, corresponding to 5 THz) can be increased by further engineering the dispersion of the metasurface, i.e., by further optimizing the position, linewidth, and dispersion of the two optical modes. In particular, larger operational bandwidths can be obtained if the spectral detuning between the two optical modes considered in Fig. 2 (see also the range indicated by $\Delta$ in Fig. 1c) is increased, provided that the quality factor of each mode is simultaneously decreased in order to keep the normal-incidence transmission low. As an example, in the Supplementary Information (Supplementary Section S6) we show numerical calculations of an additional optimized design, obtained by slightly tweaking the geometry in this work, which features a bandwidth of 10 THz, i.e., twice the one of the device considered in Figs. 2–5. Moreover, this dispersion engineering approach demonstrated here can be expanded to scenarios involving more than two optical modes, which allow a larger operational bandwidth and additional flexibility in engineering the angle-dependent response, albeit at the expense of increased design complexity.

Our results demonstrate that it is possible to design and realize simple single-layer metasurfaces to perform analog image computation in realistic scenarios where, for example, the input polarization or frequencies are unknown, and when large insertion loss cannot be tolerated. Moreover, the simplicity of the proposed design makes it amenable to mass manufacturing, an important requirement for future commercialization of these devices. We anticipate that further improvement of the proposed design can lead to even larger operational bandwidths and efficiencies. Moreover, we expect that more complex dispersion engineering, such as controlling the position and dispersion of additional zeros and poles, may lead to more sophisticated transfer functions and advanced functionalities.

## Methods

### Sample fabrication

The samples were fabricated with a standard top-down lithographic process. Glass coverslips (25 × 75 × 1 mm, Fisher Scientific) were used as transparent substrates. The substrates were cleaned by placing them in an acetone bath inside an ultrasonic cleaner, and later in an oxygen-based cleaning plasma (PVA Tepla IoN 40). After cleaning, a layer of 270 nm of amorphous silicon (α-Si) was deposited via a plasma-enhanced chemical vapor deposition (PECVD) process. A layer of E-beam resist (ZEP 520-A) was then spin-coated on top of the samples, followed by a layer of an anti-charging polymer (DisCharge, DisChem). The desired photonic crystal pattern was then written with an electron beam tool (Elionix 50 keV). After ZEP development, the pattern was transferred to the underlying silicon layer via dry etching in an ICP machine (Oxford PlasmaPro System 100). The resist mask was finally removed with a solvent (Remover PG). SEM pictures of the final device are shown in the Supplementary Information, Fig. S1.

### Optical characterization

The transmission amplitudes shown in Fig. 2g–j were performed with a custom-built setup described in more detail in the Supplementary Information (Section S1). The sample was mounted on two different rotation stages to control the polar angle $\theta$ and the azimuthal angle $\phi$. A broadband supercontinuum laser (NKT, SuperK) was filtered via a commercial narrowband filter (Photon, LLTF Contrast) and then injected into the setup via a fiber. The laser was weakly focused on the metasurface via a lens with f = 20 cm focal length. The transmitted signal was collected and re-collimated on the other side of the sample by an identical lens. Two identical germanium powermeters (Thorlabs, S122C) were used to measure the transmission level through the metasurface. A linear polarizer placed before the beamsplitter was used to polarize the incoming beam along either x or y, which correspond, respectively, to p- and s-polarization for any value of $\theta$ and $\phi$. The transmission amplitudes shown in Fig. 2g–j were then obtained by sweeping the angle $\theta$ and the input wavelength and recording the powers measured by the powermeters.

The imaging experiments shown in Figs. 4 and 5 of the main paper were performed with the setup shown in Fig. 4a. The illumination was provided by the same supercontinuum source used in the setup described in the previous paragraph. For the measurements in Fig. 4b, c, the broadband source was filtered by the same narrow band filter used for the transmission measurements. For the measurements in Fig. 5, the output of the supercontinuum laser was filtered with a custom-built pulse shaper that allows to continuously tune the linewidth and central wavelength of the input spectrum.

## Data availability

All data that support the findings of this work are shown in the main text and Supplementary Information. The experimental data generated in this study and shown in Figs. 2, 4, and 5 have been deposited in the repository https://doi.org/10.5281/zenodo.10004018.

## Code availability

All electromagnetic simulations (Figs. 2b–f and 3a–j) were performed using a commercially available electromagnetic solver (Ansys HFSS), and they can be readily reproduced by using the geometrical parameters provided in the text. The simulated image processing results in Fig. 3k–o were obtained with a custom-made Matlab script that implements numerically the formulas shown in Supplementary Information (Supplementary Section S3).

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

## Acknowledgements

This work was supported by the Air Force Office of Scientific Research MURI program and the Simons Foundation. The authors would like to thank Dr. Dmitriy Korobkin for building the pulse shaper used for the experiment discussed in Fig. 5.

## Author contributions

All authors conceived the idea and the corresponding experiment. A.Ar. performed numerical simulations and optimization, and he conducted the numerical image analysis with assistance from M.C.; M.C. fabricated the devices and performed the experimental measurements together with S.S.; A.Al. supervised the project. All authors analyzed the data and contributed to writing the manuscript.

## Competing interests

The authors declare no competing interests.
