## [Peer Review File · Nature Communications]

Reviewers' comments:

Reviewer #1 (Remarks to the Author):

I have read with interest the manuscript « Dispersion Engineered Metasurfaces for Broadband, High-NA, High-Efficiency, Dual-Polarization Analog Image Processing » by Michele Cotrufo, Akshaj Arora, Sahitya Singh and Andrea Alù that was submitted for publication in nature communications (NCOMMS-22-53712-T).

In this paper, the authors engineered the band structure of a periodic nonlocal metastructures, or simply a hole in a Si slab waveguide on a glass wafer, to modulate the transmission as a function of the incident angle. The aim of this manipulation is to achieve high transmission of large spatial frequency Fourier components, while removing the low frequency components, to realize image processing and in particular edge detection. The motivation of image processing with metasurfaces is already well established in the literature and the approach of using dispersion engineering for this purpose is also well-known.

The innovation presented in this manuscript is related to the methodology to achieve spatial frequency dependent transmission using not only one but two resonances, designed so as to spectrally shift in opposite direction as a function of the transverse momentum. Because of the symmetry of the structure considered by the authors, it also enables polarization independent response. Overall the work is relatively straightforward, and might not be sufficiently innovative to deserve high impact publication. Because of the achieved figures of merit including “broadband” (although only about 35nm), polarization-independent, high-NA and high-efficiency edge detection with responses up to NAs ~ 0.35 , this paper would deserve publication in a more specific applied photonics journal such as Applied Physics Letters or ACS Photonics.

Reviewer #2 (Remarks to the Author):

Dispersion Engineered Metasurfaces for Broadband, High-NA, High-Efficiency, Dual-Polarization Analog Image Processing by Michele Cotrufo et.al.

Optical analog computing refers to perform different mathematical operations on the incident light field. Compared with digital processing approach, it owns a lot of advantages, such as high speed, low energy consumption etc. In recent years, researchers revisit the optical analog computing due to the development of nanotechnology. Edge detection is one of the important analog computing methods, which can filter out the most important information of the object. In this work, the authors demonstrate the two-dimensional edge detection for an arbitrary polarization based on a single layer photonics

crystal metasurface. It achieves 35 nm bandwidth and 0.35 numerical apertures. However, the paper is not well written and it is also largely overstated. Although the authors have shown detailed numerical and experimental results, it is hard to find clear advantages and novelties. Also, I do not see how this approach improve the state-of-the-art techniques. Unfortunately, I cannot recommend the publication in Nature communications.

1. The authors emphasize the performance of edge detection with broadband, polarization-independent, and high-efficiency advantages. However, these performances have been widely studied in this field, which are not mentioned in this paper.

Huo, P., Zhang, C., Zhu, W., Liu, M., Zhang, S., Zhang, S., Chen, L., Lezec, H.J., Agrawal, A., Lu, Y. and Xu, T., 2020. Photonic spin-multiplexing metasurface for switchable spiral phase contrast imaging. *Nano Letters*, 20(4), pp.2791-2798.

Kim, Y., Lee, G.Y., Sung, J., Jang, J. and Lee, B., 2022. Spiral metalens for phase contrast imaging. *Advanced Functional Materials*, 32(5), p.2106050.

Zhou, J., Qian, H., Zhao, J., Tang, M., Wu, Q., Lei, M., Luo, H., Wen, S., Chen, S. and Liu, Z., 2021. Two-dimensional optical spatial differentiation and high-contrast imaging. *National science review*, 8(6), p.nwaa176.

2. The authors claimed that the metasurface designed for edge detection has a broadband wavelength, i.e., 35 nm bandwidth at the working wavelength 1500 nm. However, the edge detection based on PB phase metasurface can easily gain several hundred nanometer bandwidths.

3. The authors are overclaim that the metasurface has a high efficiency. In the main text, experimentally, the average efficiency of the metasurface is only achieved around 1%. It is well known that the metasurface can easily achieve 50%- 80% efficiency, which will be useful for practical applications.

4. The readers may curious about that the mentioned edge detection method works for incoherent light or not. Explanations should also be provided.

Response to the Referees' Reports

Reviewer #1 (Remarks to the Author):

I have read with interest the manuscript « Dispersion Engineered Metasurfaces for Broadband, High-NA, High-Efficiency, Dual-Polarization Analog Image Processing » by Michele Cotrufo, Akshaj Arora, Sahitya Singh and Andrea Alù that was submitted for publication in nature communications (NCOMMS-22-53712-T).

In this paper, the authors engineered the band structure of a periodic nonlocal metastructures, or simply a hole in a Si slab waveguide on a glass wafer, to modulate the transmission as a function of the incident angle. The aim of this manipulation is to achieve high transmission of large spatial frequency Fourier components, while removing the low frequency components, to realize image processing and in particular edge detection. The motivation of image processing with metasurfaces is already well established in the literature and the approach of using dispersion engineering for this purpose is also well-known.

The innovation presented in this manuscript is related to the methodology to achieve spatial frequency dependent transmission using not only one but two resonances, designed so as to spectrally shift in opposite direction as a function of the transverse momentum. Because of the symmetry of the structure considered by the authors, it also enables polarization independent response. Overall the work is relatively straightforward, and might not be sufficiently innovative to deserve high impact publication. Because of the achieved figures of merit including “broadband” (although only about 35nm), polarization-independent, high-NA and high-efficiency edge detection with responses up to NAs ~ 0.35 , this paper would deserve publication in a more specific applied photonics journal such as Applied Physics Letters or ACS Photonics.

We thank the reviewer for reviewing our manuscript and for providing comments. We respectfully disagree with many of the reviewer's statements, and we feel that the reviewer has, in many regards, downplayed the results of our work in a way that indicates either a very poor reading of our work, or a biased position. In the following, we provide a point-by-point answer to the reviewer's comment. For convenience, we have split the reviewer's comment in separate parts and added numbering. In light of the clarifications and additional explanations that we provide in this answer and in the revised manuscript, we hope that the reviewer will reconsider her/his assessment on our manuscript.

The motivation of image processing with metasurfaces is already well established in the literature and the approach of using dispersion engineering for this purpose is also well-known.

We agree that “*the motivation of image processing with metasurfaces is already well established*”. However, we do not think that this constitutes a valid criticism to our work. While it is true that the idea of using metasurfaces for image processing has been already discussed, our work is the first one that (1) proposes and demonstrates a general method to design a metasurface that performs image processing with metrics that are not met by any previous work, either theoretical or experimental, and (2) demonstrates experimentally that this recipe can be implemented in a single-layer ultrathin metasurface. We believe that the combination of a novel approach to achieve optimized metrics and the clear experimental demonstration warrants publication in Nature Communications.

Moreover, we strongly disagree with the reviewer's statement that “*the approach of using dispersion engineering for this purpose is also well-known*”. First, if this approach is “well-known”, we would have expected her/him to point out some relevant work in literature. The lack of any citation makes it particularly hard and unfair for us to properly address this comment. In fact, we are confident that none of the works

available in the literature discusses or implements the novel dispersion engineering recipe that we propose and demonstrate here, which relies on optimizing the position and dispersion of two different modes to achieve simultaneously broad bandwidth, polarization independent and large NAs. We want to stress in this regard that ‘dispersion engineering’ is a broad term, which is applicable to many instances. In our work, realizing a broader bandwidth edge detection requires to position very closely two zeros of the normal-incidence reflection coefficient of our device, and make sure that their angular dispersion is oppositely sloped. This is by far not a trivial task, especially considering that we have achieved this feat experimentally and within a single-layer metasurface device. Vaguely remarking that ‘dispersion engineering’ is well-known is not at all capturing the importance and relevance of our finding.

In addition, as we also mention in the manuscript, all the experimental works that use dielectric metasurfaces for edge detection in real-space (that is, without a 4f system) rely on a single resonant mode. As an example, consider the works [1] Cordaro et al, Nano Lett., vol. 19, no. 12, Art. no. 12, Dec. 2019 and [2] Zhou et al, Nat. Photonics, vol. 14, no. 5, Art. no. 5, May 2020. In these works (and other similar works) the desired Laplacian response is obtained thanks to the spectral shift of a single optical mode as the angle increases. These approaches are much easier to implement and design, but they are prone to very narrow bandwidths (as in [1]), and strong anisotropy and polarization asymmetries (as in both [1] and [2]). Our dispersion engineering approach, which to the best of our knowledge has not been discussed in any other work, is capable of optimizing all these and other metrics simultaneously.

To clarify this point, we made the following edits to the revised manuscript:

1. The text in the 5th paragraph of the Introduction reads
“A common approach [19]–[21], [24], [28] relies on inducing a single optical mode in a nonlocal metasurface, leading to a Fano lineshape in the normal-incidence transmission spectrum, [...] While this approach has proven successful to realize analog edge detection, its operational bandwidth is inherently limited.”
2. In the Discussion section we added the sentences:
“Differently from conventional methods that rely on the angle-dependent spectral shift of a single optical mode, our approach relies on accurately engineering the dispersion curves of two different modes in a transversely invariant nonlocal metasurface. We have shown that this method provides enough flexibility and degrees of freedom to simultaneously optimize all the relevant figures of merit. Despite its apparent complexity, this approach can be readily implemented in different platforms and materials. In particular, we have experimentally implemented [...]”

1. The innovation presented in this manuscript is related to the methodology to achieve spatial frequency dependent transmission using not only one but two resonances, designed so as to spectrally shift in opposite direction as a function of the transverse momentum. Because of the symmetry of the structure considered by the authors, it also enables polarization independent response.

We thank the reviewer for pointing out one of the innovations of our work. As we explained also in the previous point, we believe that this methodology is not a straightforward extension of previous works, since it provides a completely new recipe to design edge-detection metasurfaces where all relevant figures of merit are simultaneously optimized.

We are honestly appalled and strongly disagree with the reviewer’s statement that *“Because of the symmetry of the structure considered by the authors, it also enables polarization independent response.”*. This statement seems to imply that the polarization-independent response of our metasurface is a trivial

Figure R1. Response of a metasurface with the same symmetry as the one considered in this manuscript, but with different values of the geometrical parameters. All plots correspond to measured data. (a) Normal incidence transmission spectra (for both s- and p-polarized waves. (b) Transmission amplitude of s- and p-polarized waves as a function of the polar angle θ , for fixed azimuthal angle $\phi = 0^\circ$. (c-d) Full co-polarized transfer functions, for s- and p-polarized waves.

consequence of using a design with a six-fold symmetry, thus downplaying the role of our dispersion engineering approach. **This is not the case at all. A device with six-fold symmetry provides a polarization-independent response only at normal incidence, for which our devices are designed to totally reflect.** In order to perform polarization-independent edge detection, the device must retain its polarization-independence at all angles of interest. This is very far from being trivial and, as a testimony of this fact we stress that such dual-polarization functionality has not been experimentally achieved before. As an example, Figure R1 shows some results from a metasurface with **the same six-fold symmetry** as the device that we used in the submitted manuscript, but with different geometrical parameters. As clear from Figs. R1b and R1(c-d), the response of the metasurface is strongly different for s- and p-polarized waves. In particular, the s-polarized transmission curve provides the desired quadratic increase with angle, while the p-polarized transmission remains almost zero across the NA of interest. In turn, this implies that the quality of the edge detection will be strongly dependent on the polarization of the input image. We emphasize once more that the polarization-independent edge detection of our device has never been achieved in any of the previous experimental works that we are aware of, see for example (1) : Cordaro et al, Nano Lett., vol. 19, no. 12, Art. no. 12, Dec. 2019 and (2): Zhou et al, Nat. Photonics, vol. 14, no. 5, Art. no. 5, May 2020.

In order to properly address this, in the revised manuscript we modified the last part of the first paragraph of the “General Principle and Metasurface Design” section, that now reads

“[...] working with lattices with C_6 rotational symmetry leads to a polarization-independent response at normal-incidence, and it guarantees the largest possible degree of isotropy for tilted angles. However, we emphasize that a C_6 rotational symmetry does not automatically guarantee polarization-independent response at off-normal angles, which is instead crucial to achieve uniform edge detection independently of the polarization of the input image. Achieving a full-angle polarization-independent response requires to further engineer the dispersion of the metasurface to ensure that the relevant optical modes have a similar coupling to s- and p-polarized waves.”

- Overall the work is relatively straightforward, and might not be sufficiently innovative to deserve high impact publication. Because of the achieved figures of merit including “broadband” (although only about 35nm), polarization-independent, high-NA and high-efficiency edge detection with responses up to NAs ~ 0.35 , this paper would deserve publication in a more specific applied photonics journal such as Applied Physics Letters or ACS Photonics.

We respectfully disagree with the reviewer's statement that '*Overall the work is relatively straightforward*', following our responses to the previous comments. In particular, the reviewer's claim of "straightforwardness" strongly contrasts with the fact that none of the works available in literature either (1) implements the dispersion engineering recipe that we propose and demonstrate to optimize all relevant metrics of performance, or (2) provides any alternative metasurface design where all the relevant metrics are simultaneously optimized, and anywhere close to the values that we report. In particular, as we already pointed out in our answer to the previous comment, **achieving a polarization-independent response at large angles (which is fundamental to obtain polarization-independent and high-efficiency edge detection) is far from being 'straightforward'**.

Reviewer #2 (Remarks to the Author):

Optical analog computing refers to perform different mathematical operations on the incident light field. Compared with digital processing approach, it owns a lot of advantages, such as high speed, low energy consumption etc. In recent years, researchers revisit the optical analog computing due to the development of nanotechnology. Edge detection is one of the important analog computing methods, which can filter out the most important information of the object. In this work, the authors demonstrate the two-dimensional edge detection for an arbitrary polarization based on a single layer photonics crystal metasurface. It achieves 35 nm bandwidth and 0.35 numerical apertures.

We thank the reviewer for his/her work on our manuscript, and for providing comments. We respectfully disagree with some of the reviewer's statements, because we feel that they are either not adequately substantiated, or they stem from deep misunderstandings of the physics at play in our device. Here below we provide a point-by-point answer to the reviewer's comment. For convenience, we have split the reviewer's comments in separate parts and added numbering. In light of the clarifications and additional explanations that we provide in this answer and in the revised manuscript, we hope that the reviewer will reconsider her/his assessment on the novelty, impact and usefulness of our approach and experiment.

However, the paper is not well written and it is also largely overstated. Although the authors have shown detailed numerical and experimental results, it is hard to find clear advantages and novelties. Also, I do not see how this approach improve the state-of-the-art techniques. Unfortunately, I cannot recommend the publication in Nature communications.

We are sorry to hear that the reviewer finds our paper "*not well written*", however, we struggle to understand what might have caused this comment. It is not clear if the reviewer finds the structure of our paper confusing, or if he/she is complaining about style/grammar issues, it is hard to respond to criticisms that do not pinpoint specific issues. We are also puzzled by the reviewer's comment that our paper is "*largely overstated*". Such a strong statement would require, at a minimum, that the reviewer points out which claim of our paper is largely overstated and why. All the claims made in our paper are quantitative, and they are based on the measured and simulated data shown in the figures. Remarkably, in the following sentence the reviewer admits this: "*Although the authors have shown detailed numerical and experimental results, [...]*". In comment #3 (below) the reviewer suggests that we have overstated the high efficiency of our design with respect to other works. As we explain in the answer to that comment, it appears that the reviewer has totally missed what efficiency means in the context of edge detection, and is comparing totally different definitions of efficiency. In fact, in the following **we show that our edge-detection efficiency is close to the theoretical maximum achievable with a passive device.**

We also strongly disagree with the reviewer's comment that "*Also, I do not see how this approach improve the state-of-the-art techniques.*". As we clearly stated in our manuscript, our dispersion engineering approach is capable of largely improving the state-of-the-art of edge detection with metasurfaces, because it simultaneously increases all the relevant metrics. In particular, our device achieves **simultaneously** large bandwidth (>35 nm), large NAs (NA=0.35), large efficiency (see our answer to comment #3 below), complete polarization independence and the largest possible isotropy. The simultaneous optimization of these 5 figures of merit is not available in any of the experimental demonstrations available in literature, which do not come even remotely close to the performance metrics reported in this paper.

1. The authors emphasize the performance of edge detection with broadband, polarization-independent, and high-efficiency advantages. However, these performances have been widely studied in this field, which are not mentioned in this paper.

- (1) Huo, P., Zhang, C., Zhu, W., Liu, M., Zhang, S., Zhang, S., Chen, L., Lezec, H.J., Agrawal, A., Lu, Y. and Xu, T., 2020. Photonic spin-multiplexing metasurface for switchable spiral phase contrast imaging. *Nano Letters*, 20(4), pp.2791-2798.
- (2) Kim, Y., Lee, G.Y., Sung, J., Jang, J. and Lee, B., 2022. Spiral metalens for phase contrast imaging. *Advanced Functional Materials*, 32(5), p.2106050.
- (3) Zhou, J., Qian, H., Zhao, J., Tang, M., Wu, Q., Lei, M., Luo, H., Wen, S., Chen, S. and Liu, Z., 2021. Two-dimensional optical spatial differentiation and high-contrast imaging. *National science review*, 8(6), p.nwaa176.

We are very confused about the meaning of the reviewer's statement that "*these performances have been widely studied in this field*". If the reviewer means that the *importance* of these figures of merit was already discussed in literature, we potentially agree with him/her (although we are not aware of any relevant paper). However, if the reviewer means that these figures of merit were already simultaneously optimized in other works, we note that she/he fails to point out which literature we are missing. As stated in the original manuscript, **we are not aware of any theoretical or experimental demonstration of subwavelength edge-detecting metasurface devices where all these metrics** (NA, bandwidth, isotropy, polarization-independence and efficiency) **are simultaneously optimized**.

The reviewer then lists 3 papers (we have added the numbering for convenience) without providing any indication of how these papers should be compared to ours. Frankly, we struggle to find any connection with our work, nor with the criticism raised by the reviewer.

First, in paper (1) and (3) **the edge-detection experiment is performed in a 4f modality**. That is, the image is placed in the first focal plane of a lens, and the metasurface is placed in the back focal plane of the lens (see for example Fig. 3a of paper 1, and Fig. 5a of paper 3). This corresponds to the standard and well-known *4f filtering technique* (see for example chapter 4 of "Fundamental of Photonics", Saleh & Teich). Indeed, the metasurfaces discussed in papers (1) and (3) are spatially varying devices, that is, they filter the impinging plane waves selectively depending on the spatial position. This is in stark contrast to our nonlocal metasurface, which **filters plane waves with respect to angle and it does not require a 4f system to perform edge detection**. In this regard, the metasurfaces considered in paper (1) and (3) are not fundamentally different from standard spatial masks used in 4f systems. Since this approach requires a 4f system to work, its footprint is fundamentally constrained and cannot be miniaturized. **Claiming that our results are not novel because of the works in (1) and (3) would be akin to reject a paper about a metalens device because of the existence of conventional lenses**. It is obvious that our device drastically miniaturizes the footprint of the resulting system, and makes it much less sensitive to the position of the metasurface.

In paper (2), a slightly more complex approach is used. The system still relies on a spatially varying metasurface, and **a 4f system is still required to do edge detection**. However, here the authors integrate the first lens on the metasurface chip. While this results in a more compact (albeit more complex to fabricate) device, it also leads to large inefficiencies, as also mentioned in their paper (see comparison below).

Importantly, in all three papers **the devices rely on polarization conversion**, and two cross-polarizers (one in input and one in output) are additionally required. Because of this, the response of these systems is strongly polarization-dependent, that is, **only one specific input polarization will create the optimal conditions to observe edge detection**. Moreover, these additional optics make these systems even more bulky, in stark contrast to our approach which does not require any additional lens or polarization optics.

Strikingly, even neglecting the important fact that the papers mentioned by the reviewer deal with devices that work in a fundamentally different (and arguable less attractive) way than ours, we emphasize that **our metrics are still largely better than those devices**. On one hand, the devices discussed in papers (1-3) have a spectral bandwidth larger than ours. This is not surprising since working in a 4f modality effectively decouples the spectral and angular response of the metasurface. We emphasize that **similar spectral performance would be obtained in a standard k-space 4f setup, without the need of a flat metasurface**.

However, none of the devices in papers (1-3) provides a combination of large NA, large maximum transmission, and polarization-independent response comparable to ours. In detail:

- In paper 1, the device has an **NA smaller than 0.02** (see their Fig. S1) and it is **polarization dependent**. A clear value of the overall efficiency is not reported in the paper.
- In paper 2, a large metalens NA=0.8 is mentioned by the authors, but this comes at the cost of large inefficiencies. While the authors do not provide clear numbers on the output intensity, from their paper is clear that **most of the input power is lost in the process of focusing light and converting the polarization**. The main text mentions “*The measured focusing efficiencies are defined as the focused power ratio of the opposite helicity to the left circularly polarized light incident power, which are 16.34%, 20.83%, 35.03%, and 22.53%*”. We note that these efficiencies correspond only to the process of converting and focusing light on the metasurface, and not to the overall intensity of the final image (see also comment #3 below). **This device is polarization dependent**.
- In paper 3, the device has an **NA of 0.08** and a **maximum transmission (at $k_x/k_0 = 0.08$) of 6%** (see their Fig. 3f). This should be compared to our maximum transmission larger than 70% (see our Figs. 2h and 2j). This device is polarization dependent.

In conclusion, we strongly disagree with the suggestion of the reviewer that the metrics demonstrated in our work have been already achieved in previous devices. This is based on two facts. (1) The papers pointed out by the reviewer consider devices that work in a fundamentally different way, much less appealing for device miniaturization. (2) Even neglecting the previous point, the combination of metric optimization in our device largely beats most of the performance shown in these other works.

In order to properly comment on this different class of devices (and also to address the next comment of the reviewer), in the revised paper we have added a paragraph in the introduction,

“A different class of edge-detecting devices has been recently demonstrated by placing Pancharatnam-Berry phase metasurfaces in the Fourier plane of a 4F system combined with two crossed polarizers [30]–[32]. This approach typically leads to large operational spectral bandwidths, since working in a 4F modality allows to effectively decouple the spectral and angular

response of the metasurface. However, the 4F arrangement and the required polarizers strongly constrain the minimum footprint of this approach, and thus limit the possibility of miniaturization.”

where ref. 30-32 are the 3 papers mentioned by the reviewer.

2. The authors claimed that the metasurface designed for edge detection has a broadband wavelength, i.e., 35 nm bandwidth at the working wavelength 1500 nm. However, the edge detection based on PB phase metasurface can easily gain several hundred nanometer bandwidths.

The reviewer is comparing the bandwidth that we achieve in our work with the bandwidth achieved in experiments that use a “PB phase metasurface”. We assume that the reviewer is referring to the experiments shown, for example, in the papers (1-3) of comment #1. It is indeed true that in these experiments a bandwidth of hundreds of nanometers is achieved. However, as we explained in the answer to comment #1, all these experiments are performed in a 4f modality, whereby the metasurface is placed at the back focal plane of a lens, and it acts as a spatially varying filter, to filter out different Fourier components. In this regard, the large bandwidth is not surprising: when working in a 4f modality the angular and spectral response are completely decoupled. **We emphasize that the large bandwidth of these devices is not a peculiar property of the metasurfaces, but it is an intrinsic advantage of working in 4f modality, with a much bulkier system. Again, this would be like comparing the bandwidth or chromatic dispersion of a metalens to the one of a regular lens.** Very similar bandwidths (or even better) would have been achieved if a metallic mask was placed in the back focal plane of the first lens, as it commonly happens in standard 4f filtering. However, as also explained in the answer to comment #1, **working in a 4f modality requires working with much bulkier optical setups. Moreover, in the specific case of the metasurfaces in papers (1-3), the edge detection also requires an input and output polarizer in order to cross-polarize the filtered signal,** making the device even bulkier and lowering the efficiency. Thus, a comparison between our experiment and the experiments in papers (1-3) is completely misplaced, since it involves two completely different approaches and drastically different footprints.

3. The authors are overclaim that the metasurface has a high efficiency. In the main text, experimentally, the average efficiency of the metasurface is only achieved around 1%. It is well known that the metasurface can easily achieve 50%- 80% efficiency, which will be useful for practical applications.

We are again puzzled by this comment. It seems that the reviewer is misleadingly comparing the numbers that we provide in our paper for **the edge-detection efficiencies** η_{peak} or η_{avg} with some other definition of efficiency.

As we explained in our manuscript, to assess the performance of an edge-detection metasurface it is important to quantify how the intensity of the final filtered image compares with intensity of the input image. This metric is what we refer to when talking about **efficiency**, because from a practical point of view it is the metric of interest for a user. Indeed, this efficiency dictates, for example, how longer the camera should integrate, or how more powerful should the illumination be, in order to have the same signal level in the output and input image. We also note (both here and in the manuscript) that this metric has never been quantitatively assessed in previous works on edge-detection metasurfaces. That is, in all previous works (including the 3 papers mentioned by the reviewer in comment #1) **the 2D plots showing the input and output images are always displayed in arbitrary units.** Our work is the first one to even define these metrics.

In our manuscript we introduced two different metrics: η_{peak} is the highest intensity in the output image divided by the highest intensity in the input image. η_{avg} , instead, is the intensity of the output image averaged in a small spatial region around the edges, divided by the highest intensity in the input image. In

Figure R2. Calculated image processing assuming an ideal polarization-independent filter. (a) Co-polarized ideal transfer functions $t_{ss}(k_x, k_y) = t_{pp}(k_x, k_y)$ of the ideal filter (see text for details). (b) Input image. The shape and dimensions are almost identical to the target used in the experiments in Figs. 4 and 5 of the main text. (c) Output image calculated assuming the transfer function in panel a. (d) Numerically calculated Laplacian of the input image.

our experiments, η_{peak} is as high as 3.5% for broadband illumination and about 5%-10% for monochromatic illumination, while $\eta_{avg} \geq 1\%$ in both cases. As we discuss in a new section of the supplementary material (section S5), while these values might “sound” small, they are actually **very close to the theoretical maximum efficiency that can be achieved with any k-space passive filtering device for a given NA**. This is summarized in Fig. R2 here below (corresponding to the new Fig. S7 in the SM). We assume an ideal k-space filter (Fig. R2a) with the exact Laplacian response up to an NA = 0.35, and with constant unity transmission for larger k-vectors. We then consider an input image (Fig. R2b) identical in shape and size to the one considered in the experiments, and we calculate the expected output image (Fig. R2c) upon the action of the ideal filter in Fig. R2a. The peak intensity in the filtered image is 8%-9%. These values match very well the typical values of 5%-10% obtained experimentally for monochromatic excitation (Fig. 4 of main paper). This shows that, for a given NA, **the efficiency of our device is essentially the same as the one of an ideal edge-detector**. Moreover, in Fig. R2d we show the image obtained by applying the exact Laplacian operator to the same input image. The peak intensity in Fig. R2d is about 20% of the input intensity. Thus, **the experimentally measured peak efficiency $\eta_{peak} \geq 5\%$ of our device is actually quite close to the upper bound dictated by the intrinsic properties of the desired mathematical operation**. The fact that the efficiencies are nominally low should not be surprising: these devices filter out low-momenta impinging on the metasurface, and transmit only the intensity localized at the edges of the image. It is not surprising that the transmission efficiency is therefore far from unity.

We guess that the “50%- 80%” efficiency mentioned by the reviewer refers to the maximum transmission of a metasurface at a given angle. For edge detection, **the metasurface needs to be strongly non-transmissive at normal incidence, thus having high transmission at $\theta = 0$ would actually be deleterious**. Large transmission values are instead desired at large angles, and indeed our metasurface provides transmission values larger than 50% at $\theta > 20^\circ$ (see Figs. 2h and 2j). These large transmission values are one of the key ingredients (together with the polarization-independence) that allow us to achieve high-efficiency edge detection.

In conclusion, we believe that the reviewer’s statement that our efficiency is much lower than other approaches stems from a fundamental misunderstanding of how the efficiency of edge-detection metasurfaces is quantified. In fact, **we have shown that the edge-detection efficiency of our metasurface is close to the maximum value allowed for any passive k-space filtering.**

This misunderstanding was possibly due to a lack of proper explanation from our side, which we addressed in the revised manuscript. In particular,

- We modified the last paragraph of the “General Principle and Metasurface Design” section to better explain the difference between different efficiency metrics:

In order to quantitatively assess the efficiency of our edge-detecting metasurface, it is necessary to quantify how the intensity of the filtered image compares with the intensity of the input image. This metric is of crucial importance for the implementation of these devices in real-world applications because it dictates, for example, how much the integration time of a camera needs to be increased in order to have the same level of noise in the output and input image. To this aim, we introduce two different metrics.

[...]

Importantly, we show in [30] that the values of $\eta_{peak}=7\%-9\%$ found in Fig. 3 are very close to value of η_{peak} obtainable with the best possible k-space filtering device with a fixed $NA = 0.35$, and they are only a factor of $\sim 2x$ smaller than what would be obtained by applying the exact mathematical Laplacian operator on the input image. As we show in the next section, comparable values of η_{peak} are also obtained in the experiments. Thus, the efficiency our metasurface is close to the maximum value obtainable with any passive k-space filtering devices performing a Laplacian operation for a given NA. We emphasize that such high efficiencies are due to the fact that our metasurface provides a Laplacian-like response for both polarizations, and with large values of transmission amplitudes at large angles.

- For the sake of clarity, we made some changes to the simulations results shown in the bottom line of Fig. 3. In the previous version of the manuscript the simulations were done assuming a CUNY logo with width $\sim 350\lambda \approx 525 \mu m$. This is larger than the image size that we considered in the experiment, $\sim 100\mu m$. In order to be able to properly compare the efficiency between the simulations and experiments, we replaced the bottom line of Fig. 3 with simulations corresponding to CUNY logos with approximately the same size as in the experiments.
- We added a new section (section S5) to the supplementary materials to show that the peak efficiency that we achieve in our experiment are actually very close **to the maximum efficiencies achievable with any ideal image-processing based on k-space filtering.**

4. The readers may curious about that the mentioned edge detection method works for incoherent light or not. Explanations should also be provided.

We thank the reviewer for this interesting comment. Indeed, this edge detection method requires that the illumination is at least partially coherent, that is, that the coherence length of the illumination is larger than both the typical size of the image and the difference the optical paths travelled by plane waves corresponding to different wave vectors. We note that **this requirement is not peculiar to our method**, but it affects any optical system, including the standard 4f filtering technique. A few recent papers in

literature (<https://pubs.acs.org/doi/10.1021/acsp Photonics.9b01465>), <https://onlinelibrary.wiley.com/doi/full/10.1002/lpor.202200038>) have discussed possible routes to perform edge detection with incoherent light, although at the expense of more bulky setups and with the need of digital post-processing.

To properly comment on this, we have added the following text to the Introduction section,

“As for any analog optical filtering technique that relies on coherent superposition of different plane waves, this approach to edge detection requires some degree of spatial coherence in the illumination. We note that recent works have proposed a way to overcome this limitation [33], [34], albeit at the expense of increased footprint and the need of digital postprocessing.”

REVIEWER COMMENTS

Reviewer #1 (Remarks to the Author):

The authors are correct saying that they report for the first time the use of two resonances to control the dispersion, instead of exploiting the angular response of nanophotonic surfaces operating in the close proximity of only one resonance, as traditionally done in various papers (mostly ED resonance) see, <https://doi.org/10.1073/pnas.1820636116> ; <https://pubs.acs.org/doi/10.1021/acsp Photonics.0c01874> ; <https://opg.optica.org/ol/fulltext.cfm?uri=ol-45-7-2070&id=429562>; [28] J. Valentine's paper, and so forth.

As I stated in my previous report (it seems that it is also the only argument of innovation in the author's rebuttal letter), the bit of innovation comes from the utilization of these two resonances, as stated "very precisely to have two zeros of the reflection coefficient at normal incidence and making sure that the dispersion move in opposition". I agree with the authors that it is not a trivial affair. Generally speaking, designing systems with increasing number of constraints or physical parameters does not help reducing the complexity and the applicability of the solution.

When the authors argue that the metric are unprecedented, I suggest to read the following article: <https://www.degruyter.com/document/doi/10.1515/nanoph-2021-0239/html?lang=en>. Impressive angular performances, beyond the reported results have been obtained using an interesting edge detection approach by treating the light field images. Experimentally demonstrated edge detection is achieved in 3D with a 260nm bandwidth at visible wavelength. This paper is not even cited by the authors.

Regarding the comment on the symmetry of the nanostructure, yes the authors correctly point out that polarization independent response with six-fold symmetry occurs generally for normal incidence only. Their structure can handle polarization independent response over a larger angle range. I believe that the text added better highlights this specificity.

That being said, these results are not significant, from the metrics point of view (see comment on bandwidth and 3D edge detection above), and from the innovation on image processing, to deserve high impact publication. This work is nice but incremental, in the line of previous edge detection papers.

Reviewer #4 (Remarks to the Author):

Fast and reliable edge detection of an object is one of the fundamental capabilities required in applications such as imaging identification, machine learning, and artificial intelligence. Analog computing with optical metasurfaces holds a great potential for increasing processing speeds and reducing power consumption in edge detection. In this manuscript, the authors exploit dispersion engineering to design and realize metasurfaces capable of performing isotropic 2D edge detection. I believe some extensive revision would be needed to describe the technique and demonstrate its value more clearly. There are several issues need to be considered carefully by the authors.

1.The dispersion of the two modes is strong enough and their overall spectral shifts are larger than their linewidths. The transmission within the band will rise to almost-unitary values for large angles. This property ensures high transmission of large spatial frequency Fourier components, resulting into edge-enhanced in the output images. To perform the operation on an arbitrarily polarized input 2D image, a metasurface with a polarization independent and isotropic transfer function is necessary. Therefore, Equation 1 is one of important result in this manuscript. A polarization-independent response can be expected because of the symmetry of the structure. However, it is not clear how to obtain it from the theoretical derivation. If it is a well-established result, the corresponding literature should be cited. Therefore, the lack of clarity in the description makes it difficult to judge the novelty of the proposed scheme.

2.The authors exploit dispersion engineering to design and realize metasurfaces capable of performing isotropic 2D edge detection over a broad operational bandwidth and for any input polarization, while simultaneously maintaining high numerical aperture and record efficiency. From Figure 5, it shows that the band is only about 35nm. It is known that the isotropic 2D broadband edge detection has been reported in some literatures, for example, references [30-32] cited in this paper. In these reported results, the bandwidth of edge detection is larger than 150nm, although not all metrics are simultaneously optimized. I suggest the authors clarify the mechanism of broadband and how to improve the bandwidth of the edge detection.

3. Figure 4 experimentally verifies the quality of the edge detection under a broad-band input. The authors claim that high-quality edge detection can be obtained for any wavelength between 1452nm and 1485 nm. However, some noise peaks are also present as shown in Figure 4 (c), which may lead to the misidentification of objects, which make the superiority of the proposed method doubtful. I suggest the authors clarify the reason of noise peaks and how to eliminate them.

Response to the Referees' Comments

Reviewer #1 (Remarks to the Author):

(Note: we have added numbering for clarity)

1) The authors are correct saying that they report for the first time the use of two resonances to control the dispersion, instead of exploiting the angular response of nanophotonic surfaces operating in the close proximity of only one resonance, as traditionally done in various papers (mostly ED resonance) see, <https://doi.org/10.1073/pnas.1820636116> ; <https://pubs.acs.org/doi/10.1021/acsp Photonics.0c01874> ; <https://opg.optica.org/ol/fulltext.cfm?uri=ol-45-7-2070&id=429562>; [28] J. Valentine's paper, and so forth.

As I stated in my previous report (it seems that it is also the only argument of innovation in the author's rebuttal letter), the bit of innovation comes from the utilization of these two resonances, as stated "very precisely to have two zeros of the reflection coefficient at normal incidence and making sure that the dispersion move in opposition". I agree with the authors that it is not a trivial affair. Generally speaking, designing systems with increasing number of constraints or physical parameters does not help reducing the complexity and the applicability of the solution.

We appreciate that the reviewer acknowledges the innovation of our work. However, we disagree with the reviewer's classification that "*the bit of innovation comes from the utilization of these two resonances*". This seems to imply that simply using two resonances (rather than one) is enough to provide all the simultaneously improved metrics that we demonstrate in our work. Instead, as we believe to have clearly explained in our paper, our recipe relies on a sophisticated dispersion engineering approach, which requires controlling both position and dispersion of two independent and suitably tailored resonances.

We are puzzled by the reviewer's comment that "*designing systems with increasing number of constraints or physical parameters does not help reducing the complexity and the applicability of the solution.*" Our work provides a novel recipe to design the metasurface, and an experimental demonstration of the proposed recipe within a remarkably simple design. The experimentally demonstrated metasurface design relies on three parameters (in fact, the free parameters become two once the geometry is scaled to target a desired wavelength range). Thus, the "*increasing number of physical parameters*" definition used by the reviewer is far from being justified.

In fact, we believe that our work represents a nice synthesis between, on one hand, providing a universal recipe (whose implementation complexity will eventually depend on the specific platform) and, on the other hand, demonstrating that this seemingly complex recipe can actually be implemented in a simple platform.

The reviewer mentions previous works (<https://doi.org/10.1073/pnas.1820636116> ; <https://pubs.acs.org/doi/10.1021/acsp Photonics.0c01874> ; <https://opg.optica.org/ol/fulltext.cfm?uri=ol-45-7-2070&id=429562>; [28] J. Valentine's paper) as examples of papers that use only one resonance. We would like to emphasize again that the novelty of our work is not merely the increased number of resonances, but rather their elegant engineering to simultaneously optimize all relevant figures of merit for the proposed device. In this context, we need to emphasize that one of the works mentioned by the reviewer

(<https://doi.org/10.1073/pnas.1820636116>) performs edge detection in a 4f system and with spatially-varying metasurfaces. This is a very different setup, both conceptually and practically, from the one proposed in our work. In fact, we had already provided a detailed comparison and explanation of this in our answer to reviewer 2 in the previous rebuttal letter. In short, working in a 4f modality removes any practical advantage of using flat metasurfaces, since the overall structure needs to be four focal lengths long, and alignment issues become important. As we pointed out in the previous round of review, it would be like comparing the performance of an ultrathin metalens with the one of a commercial bulky lens thousands of wavelengths thick.

2) When the authors argue that the metric are unprecedented, I suggest to read the following article: <https://www.degruyter.com/document/doi/10.1515/nanoph-2021-0239/html?lang=en>. Impressive angular performances, beyond the reported results have been obtained using an interesting edge detection approach by treating the light field images. Experimentally demonstrated edge detection is achieved in 3D with a 260nm bandwidth at visible wavelength. This paper is not even cited by the authors.

We are puzzled by this comment. In this article, edge detection is achieved with an **electronic-based, active, and digital system**, which is totally un-related and disconnected from the approach used in our work (which relies on a passive, analog, and electronic-free and bias-free method). We do not understand the series of comments and comparisons that this reviewer continues to bring up in the various rounds of reviews, which appears to be totally irrelevant and unconstructive to the review of our work.

In the work mentioned by the referee, a system of metalenses is used to rely an optical image on a camera. **The full unfiltered image is relied to the camera.** The unfiltered image is then acquired and digitalized through the camera, and the edge detection is performed digitally by subtracting the intensities of neighboring pixels. This is very clear from Fig. 1 of the manuscript and from the accompanying text (see below), which makes us wonder how the reviewer could miss this. This approach could be equally implemented with a system of two standard lenses, without requiring any meta-optics at all. The only difference between a standard two-lens 4f imaging system and the system considered in this reference is that, in the latter case, the second lens consists of an array of several smaller metalenses. It appears that the authors have done so in order to digitalize the image even before it reaches the camera. Several portions of the paper mentioned by the referee explain clearly that, there, the edge detection is performed digitally and with the use of computational algorithms. For example:

- *“After capturing the light field raw image, **one-pixel misplaced self-subtraction along x-direction enables 1D edge information extraction**. Similarly, the 2D edge image could be realized with misplaced self-subtraction along both x and y-direction.”*
- *“The light-field image is obtained by the CMOS sensor of the commercial camera.”*
- *“The light-field raw image collected through the achromatic meta-lens array by CMOS sensor displays 56 by 38 sub-images, as shown in Figure 3a. The disparity of each sub-image of the objects is derived through the surrounding sub-images with **a brightness comparison**. The depths are calculated using the Euclidean geometry and the disparity. “*
- *“**Circle-moving the light-field raw sub-images with one pixel along x or y-direction, then subtract it with the original light-field raw image. The 1D edge information extraction is enabled by displacement of the self-subtraction.**”*

We emphasize once more that we do not find any relevance between this reference and our work. Performing edge detection digitally gives up on the main advantages of analog image processing using metasurfaces, in particular in the context of energy, time, and footprint savings.

3) Regarding the comment on the symmetry of the nanostructure, yes the authors correctly point out that polarization independent response with six-fold symmetry occurs generally for normal incidence only. Their structure can handle polarization independent response over a larger angle range. I believe that the text added better highlights this specificity.

We are glad to see that the reviewer agrees with us that achieving polarization independence is far from being a trivial task. We want to emphasize once more the importance of this aspect. Indeed, **none** of the works available in literature (including the ones mentioned by the reviewer in comment #1) is able to achieve polarization independence for a large NA.

4) That being said, these results are not significant, from the metrics point of view (see comment on bandwidth and 3D edge detection above), and from the innovation on image processing, to deserve high impact publication. This work is nice but incremental, in the line of previous edge detection papers.

We strongly disagree with the reviewer. The reviewer bases his/her claims on a comparison with the paper mentioned in his/her comment #2 above. However, as we explained above, this paper is unrelated to our work, both from a fundamental and conceptual point, and from an implementation point of view. We emphasize that in both rounds of review the referee continues to state that our results are not significant from a metric point of view. Yet the reviewer has repeatedly failed to show us any work where passive and bias-free edge-detection is performed with metasurfaces (without the need for a 4f system), and with simultaneously optimized metrics that come even remotely close to the ones that we report in our work.

Reviewer #4 (Remarks to the Author):

Fast and reliable edge detection of an object is one of the fundamental capabilities required in applications such as imaging identification, machine learning, and artificial intelligence. Analog computing with optical metasurfaces holds a great potential for increasing processing speeds and reducing power consumption in edge detection. In this manuscript, the authors exploit dispersion engineering to design and realize metasurfaces capable of performing isotropic 2D edge detection. I believe some extensive revision would be needed to describe the technique and demonstrate its value more clearly. There are several issues need to be considered carefully by the authors.

We thank the reviewer for his/her work on our manuscript, and for providing constructive criticisms. In the following, we address all the reviewer's remarks.

1. The dispersion of the two modes is strong enough and their overall spectral shifts are larger than their linewidths. The transmission within the band will rise to almost-unitary values for large angles. This property ensures high transmission of large spatial frequency Fourier components, resulting into edge-enhanced in the output images. To perform the operation on an arbitrarily polarized input 2D image, a metasurface with a polarization independent and isotropic transfer function is necessary. Therefore, Equation 1 is one of important result in this manuscript. A polarization-independent response can be expected because of the symmetry of the structure. However, it is not clear how to obtain it from the theoretical derivation. If it is a well-established result, the corresponding literature should be cited. Therefore, the lack of clarity in the description makes it difficult to judge the novelty of the proposed scheme.

We are unsure whether the reviewer is asking us to explain better why equation 1 leads to polarization-independent edge detection, or to explain how the proposed design achieves polarization independence. In the following, we address both potential questions.

The theoretical derivation of Eq. 1, shown in the SM, can be extended to show that, when equation 1 holds, polarization-independent edge detection is indeed always achieved. To show this, inspired by this comment, we have extended the derivation in section S4, with the new equations S8-S11. Moreover, Eq. 1 was also used in the theoretical paper in ref. 19. We have added a reference to [19] right before Eq. 1 in the main text.

We now discuss the second aspect, i.e., the issue of designing a metasurface with polarization-independent response. First, we would like to emphasize that, differently from what the reviewer states, the symmetry of the structure (i.e., the rotational C6 symmetry) is far from enough to guarantee polarization-independent response within the full numerical aperture, that is, for large tilted angles (see also our answer to comment 3 of Rev. 1 in this letter, and also our answer to the same reviewer in the previous round of review). The C6 rotational symmetry **only guarantees polarization independence at normal incidence**. In general, there is not a straightforward way (i.e., only based on symmetry) to obtain a polarization-independent response within a full range of tilted excitation directions. This is also the reason why polarization-independent operation has been, so far, lacking in edge-detection metasurfaces, and it is in fact one of the challenges that we addressed and overcome in our design.

We are grateful to the reviewer for pointing out that this aspect of the novelty of our design was not properly emphasized in the manuscript. To address this, we write in page 6 of the revised manuscript:

We emphasize that a C6 rotational symmetry does not automatically guarantee polarization-independent response at off-normal angles, which is instead crucial to achieve uniform edge detection independently of the polarization of the input image. Achieving a full-angle polarization-independent response requires to further engineer the dispersion of the metasurface to ensure that the relevant optical modes have a similar coupling to s- and p-polarized waves.

2. The authors exploit dispersion engineering to design and realize metasurfaces capable of performing isotropic 2D edge detection over a broad operational bandwidth and for any input polarization, while simultaneously maintaining high numerical aperture and record efficiency. From Figure 5, it shows that the band is only about 35nm. It is known that the isotropic 2D broadband edge detection has been reported in some literatures, for example, references [30-32] cited in this paper. In these reported results, the bandwidth of edge detection is larger than 150nm, although not all metrics are simultaneously optimized. I suggest the authors clarify the mechanism of broadband and how to improve the bandwidth of the edge detection.

The reviewer compares our bandwidth with the one obtained in references [30-32]. We would like to point out that these references were added during the previous round of review in order to address a similar comment from Rev. #2. As we explained both in the previous rebuttal letter and in the manuscript (see bottom of page 4), the works in Refs. 30-32 rely on a very different mechanism compared to our work, both from a conceptual and practical perspective. In these works, **edge-detection is performed in a 4f modality**. That is, the image is placed in the front focal plane of a first lens, and the metasurface is placed in the back focal plane of the first lens (see for example Fig. 3a of ref. 30, and Fig. 5a of ref. 31). This corresponds to the standard and well-known 4f filtering technique based on Fourier optics (see for example chapter 4 of “Fundamental of Photonics”, Saleh & Teich). Indeed, the metasurfaces in refs. 30-32 are spatially varying devices, that is, **they filter the impinging plane waves selectively depending on the spatial position**. This is in stark contrast to our nonlocal metasurface, which filters plane waves with respect to the angle, and it does not require a 4f system to perform edge detection. In this regard, the metasurfaces considered in refs. 30-32 are not fundamentally different from standard semi-opaque spatial masks used in 4f systems. Since this approach requires a 4f system to work, its footprint is fundamentally large and cannot be

miniaturized. Additionally, the devices in refs. 30-32 rely on polarization conversion, and two crossed polarizers (one in input and one in output) are additionally required. Because of this, the response of these systems is strongly polarization-dependent, that is, only one specific input polarization will create the optimal conditions to observe edge detection. Moreover, these additional optics make these systems even more bulky and prone to alignment issues, in stark contrast to our approach which does not require any additional lens or polarization optics.

The fact that the experiments in refs. 30-32 achieve a large spectral bandwidth is not surprising, since this advantage is inherited from the 4f configuration. We emphasize that even larger spectral bandwidths would be achieved in their works if a simpler metallic/semi-opaque mask was used in the Fourier plane instead of metasurfaces.

The stark contrast between the 4f approach used in refs. 30-32 and the one adopted in our paper is explained at page 4 of the revised paper:

A different class of edge-detection devices has been recently demonstrated by placing Pancharatnam-Berry phase metasurfaces in the Fourier plane of a 4F system combined with two crossed polarizers [30]–[32]. This approach typically leads to large operational spectral bandwidths, since working in a 4F modality allows to effectively decouple the spectral and angular response of the metasurface. However, the 4F arrangement and the required polarizers strongly constrain the minimum footprint of this approach, and thus limit the possibility of miniaturization.

3. Figure 4 experimentally verifies the quality of the edge detection under a broad-band input. The authors claim that high-quality edge detection can be obtained for any wavelength between 1452nm and 1485 nm. However, some noise peaks are also present as shown in Figure 4 (c), which may lead to the misidentification of objects, which make the superiority of the proposed method doubtful. I suggest the authors clarify the reason of noise peaks and how to eliminate them.

We thank the reviewer for his/her careful analysis of our experimental data. First, we would like to clarify that Figure 4 shows the quality of edge detection for **monochromatic inputs of different wavelengths**, while the experiment for simultaneous broadband input is shown in Fig. 5.

As the reviewer pointed out, some additional weaker peaks are visible in the experimental plots in Fig. 4c. As described in detail below, most of these peaks are not actually ‘noise’, and their presence is expected because edge-detection is implemented here via the mathematical operation of Laplacian differentiation, i.e., second-order derivative. Moreover, some other weak non-zero signals are due to smaller nonidealities of our experiment. Below, we comment on the origin of these different additional signals.

Our input images contain some “strong edges”, i.e., spatial regions where the optical intensity suddenly varies from low to high values, which lead to large absolute values of the second order derivatives and thus to large-intensity peaks in the output images. In particular, **each strong edge in the input image will result in two peaks in the output image**, as expected from the second order differentiation of a step-like function. In a practical scenario, the presence of two peaks (instead of one) for each edge does not introduce any detrimental effect – in fact, it can be used to find the exact spatial position of each edge even more accurately.

Moreover, in our input images the intensity is not perfectly flat within the bright areas. In Fig. R1 below we reproduced some of the panels of Fig. 4, together with additional cross-sectional cuts. Weak spatial fluctuations of the intensity are clearly visible both in the 2D plot in Figs. R1(a-b), and in the 1D cuts (blue

lines) in Figs. R1d and R1f, denoted by the blue arrows. These spatial intensity fluctuations are due to diffraction of light at the metallic apertures that are used in our experiment to generate the input image. Since our metasurface performs a mathematical differentiation on the whole input image, these weaker spatial variations will be differentiated as well, resulting in a set of weaker peaks in the output image. Some examples of these peaks are indicated by the orange arrows in Fig. R1(d,f).

Figure R1. (a) Input image used in the experiment in Fig. 4 of the main text. (b) Zoomed-in portion of panel a, showing the weak intensity fluctuations within the bright areas. (c) Filtered image for an input wavelength of 1455 nm. (d) Horizontal cross-sectional cut of the input image (blue lines) and output image (orange lines), corresponding to the horizontal dashed lines in panels a and b. (e-f) Same as in panels (c-d) but for a wavelength of 1470 nm.

From a practical point of view, the presence of these additional weaker peaks does not limit the capability of detecting the position of the “strong edges”, since the intensity of each peak in the output image is always proportional to the spatial derivative (at the same position) of the input image. Thus, the sharpest edges will always correspond to the two strongest peaks. In fact, the fact that we can experimentally detect such weak intensity fluctuations is a further demonstration of the large quality and efficiency of our metasurface.

Besides the effect discussed above, other unwanted additional signals (which do not necessarily have a peak-like structure) are due to a combination of the noise in the camera used for the experiment, and of the fact that the normal-incidence transmission of the metasurface, while remaining smaller than 1%, is not exactly zero. The latter effect implies that a very small portion of the ‘DC component’ of the input image is transferred into the output image without any differentiation. However, as clear from the experimental data in Figs. R1(d,f), this effect is very small, and it does not introduce any practical issue in the edge detection.

In order to clarify these issues, we have added the following text at page 12

As expected from the second-order differentiation of a step-like function, each edge results in a pair of high-intensity peaks in the output images. From a practical point of view, the presence of these two peaks (as opposed to a single peak) can be useful to better identify the spatial position of the detected edge. Besides the two high-intensity peaks, some additional weaker peaks and noise are visible in the output images (see also Fig. 4c). As explained in more detail in [35], the additional weaker peaks are due to the fact that, in the input image, the intensity is not exactly constant inside (or outside) our institution logo. Instead, weak intensity fluctuations are present due to diffraction at the metallic aperture used to create the input image (see also Fig. S8 in [35]). Since our metasurface performs high-efficiency differentiation on the whole input

image, these weaker spatial variations in the input image will be differentiated as well, resulting in additional sets of weaker peaks in the output image. From a practical point of view, the presence of these additional weaker peaks does not limit the capability of detecting the position of the main edges of the image, because the intensity of each peak in the output image remains proportional to the spatial derivative (at the same position) of the input image. In our experiment, the peaks in the output images corresponding to the main edges are about 4x more intense than the peaks corresponding to the weak intensity fluctuations.

Moreover, we added a new section to the SM (Sec. S.8), which includes Fig. S1, to explain in more details these effects.

Summary of Changes

- In order to address comment #1 of Reviewer 4, we have expanded the theoretical derivation in section S.4 of the SM, including new equations S8-S11. Moreover, we also revised the text at page 6, which reads

We emphasize that a C6 rotational symmetry does not automatically guarantee polarization-independent response at off-normal angles, which is instead crucial to achieve uniform edge detection independently of the polarization of the input image. Achieving a full-angle polarization-independent response requires to further engineer the dispersion of the metasurface to ensure that the relevant optical modes have a similar coupling to s- and p-polarized waves.

- In order to address comment #3 of Reviewer 4, we added the following text at page 12,

As expected from the second-order differentiation of a step-like function, each edge results in a pair of high-intensity peaks in the output images. From a practical point of view, the presence of these two peaks (as opposed to a single peak) can be useful to better identify the spatial position of the detected edge. Besides the two high-intensity peaks, some additional weaker peaks and noise are visible in the output images (see also Fig. 4c). As explained in more detail in [35], the additional weaker peaks are due to the fact that, in the input image, the intensity is not exactly constant inside (or outside) the CUNY logo. Instead, weak intensity fluctuations are present due to diffraction at the metallic aperture used to create the input image (see also Fig. S8 in [35]). Since our metasurface performs high-efficiency differentiation on the whole input image, these spatial intensity fluctuations in the input image will be differentiated as well, resulting in additional sets of weaker peaks in the output image. From a practical point of view, the presence of these additional weaker peaks does not limit the capability of detecting the position of the main edges of the image, because the intensity of each peak in the output image remains proportional to the spatial derivative (at the same position) of the input image. In our experiment, the peaks in the output images corresponding to the main edges are about 4x more intense than the peaks corresponding to the weak intensity fluctuations.

- In order to address comment #3 of Reviewer 4, we also added a new section to the SM (Sec. S.6), including the new figure S8.